# Identification of cuproptosis and ferroptosis-related subtypes and development of a prognostic signature in colon cancer

Yinghao He[1☯], Fuqiang Liu[1,2☯], Qingshu Li[3,4,5☯], Zheng Jiang[1]*

1 Department of Gastroenterology, The First Affiliated Hospital of Chongqing Medical University, Chongqing, China, 2 Department of Gastroenterology, The People's Hospital of Jianyang City, Jianyang, Sichuan Province, China, 3 Department of Pathology, College of Basic Medicine, Chongqing Medical University, Chongqing, China, 4 Molecular Medicine Diagnostic and Testing Center, Chongqing Medical University, Chongqing, China, 5 Department of Pathology, the First Affiliated Hospital of Chongqing Medical University, Chongqing, China

☯ These authors contributed equally to this work.
* jiangz1753@tom.com

**Data Availability Statement:** The RNA expression profiles and corresponding clinical information used in this study are available in the TCGA(https://portal.gdc.cancer.gov) and GEO(https://www.ncbi.

## Abstract

Colon cancer, as a highly prevalent malignant tumor globally, poses a significant threat to human health. In recent years, ferroptosis and cuproptosis, as two novel forms of cell death, have attracted widespread attention for their potential roles in the development and treatment of colon cancer. However, the investigation into the subtypes and their impact on the survival of colon cancer patients remains understudied. In this study, utilizing data from TCGA and GEO databases, we examined the expression differences of ferroptosis and cuproptosis-related genes in colon cancer and identified two subtypes. Through functional analysis and bioinformatics methods, we elucidated pathway differences and biological characteristics between these two subtypes. By leveraging differential genes between the two subtypes, we constructed a prognostic model using univariate Cox regression and multivariate Cox regression analysis as well as LASSO regression analysis. Further survival analysis and receiver operating characteristic curve analysis demonstrated the model's high accuracy. To enhance its clinical utility, we evaluated the clinical significance of the model and constructed a nomogram, significantly improving the predictive ability of the model and providing a new tool for prognostic assessment of colon cancer patients. Subsequently, through immune-related analysis, we revealed differences in immune cell infiltration and immune function between high- and low-risk groups. Further analysis of the relationship between the model and immune cells and functions revealed potential therapeutic targets. Drug sensitivity analysis revealed associations between the expression of model-related genes and drug sensitivity, suggesting their involvement in tumor resistance through certain mechanisms. AZD8055_1059, Bortezomib_1191, Dihydrorotenone_1827, and MG-132_1862 were more sensitive in the high-risk group. Finally, we analyzed differential expression of model-related genes between tumor tissues and normal tissues, validated through real-time quantitative PCR and immunohistochemistry. In summary, our study provides a relatively accurate

nlm.nih.gov/geo/query/acc.cgi?acc=GSE39582)
databases. All relevant data are within the
manuscript and its Supporting Information files.
The code is available at Zenodo: https://doi.org/10.
5281/zenodo.12516394.

**Funding:** The author(s) received no specific
funding for this work.

**Competing interests:** The authors have declared
that no competing interests exist.

prognostic tool for colon cancer patients, offering guidance for treatment selection and indicating the potential of immunotherapy in colon cancer.

## Introduction

As one of the most prevalent malignancies in humans, colon cancer exhibits exceedingly high incidence and mortality rates worldwide [1]. According to data from 2023, colon cancer ranks among the top three cancers in both incidence and mortality in the United States [2]. Similarly, in China, one of the most populous countries globally, colon cancer emerges as one of the predominant malignant tumors, as indicated by survey results from 2016 [3]. Despite the availability of various treatment modalities for colon cancer, including endoscopic therapy for early-stage cases, surgical intervention, adjuvant chemotherapy for metastatic disease, and immunotherapy with biologic agents, significant variations in treatment outcomes persist among patients with similar tumor staging [4, 5]. This phenomenon may be attributed to the presence of intratumoral heterogeneity, signifying the coexistence of diverse morphological, inflammatory, genetic, or transcriptomic subclones within the same tumor. Such intratumoral heterogeneity exerts notable influences on disease outcomes and treatment responses [6]. Consequently, there is a pressing need to identify additional prognostic markers and construct reliable predictive models to guide therapeutic strategies for colon cancer, thereby enhancing the survival duration of patients afflicted with this disease.

Ferroptosis, unlike currently known modes of cell death, is a regulatory form of cell necrosis induced by iron ions and reactive oxygen species-mediated lipid peroxidation [7], regulated by factors such as GPX4, FSP1, and the energy-sensing kinase AMPK [8]. Recent studies have implicated ferroptosis in various cancers including hepatocellular carcinoma, colorectal cancer, and breast cancer [9]. In colorectal cancer, ferroptosis may play a role in therapy, as several potential molecular targets with therapeutic applications have been identified in colorectal cancer models, such as GPX4, SLC 7A11, ACSL 4, and TP 53 [10]. This suggests that ferroptosis-related genes may serve as reliable prognostic markers for colon cancer.

Cuproptosis was established as a novel form of cell death in 2022. It occurs through direct binding of copper to lipoylated components of the tricarboxylic acid cycle (TCA), leading to aggregation of lipoylated proteins and loss of iron-sulfur cluster proteins, triggering protein toxicity stress, and ultimately resulting in cell death [11]. Currently, numerous research teams have conducted bioinformatics analyses on cuproptosis, suggesting its correlation with the pathogenesis of various diseases such as Crohn's disease [12], osteoporosis [13], cardiovascular diseases [14], and tumors [15]. Moreover, copper metabolism is considered a promising strategy for cancer therapy [16]. Furthermore, there is certain crosstalk between ferroptosis and cuproptosis [17]. Mitochondria, as a critical site for both ferroptosis and cuproptosis, have the TCA cycle as their convergence point. On one hand, the mitochondrial TCA cycle and electron transport chain play a central role in initiating ferroptosis by promoting mitochondrial membrane potential hyperpolarization and the accumulation of lipid peroxides [18]. On the other hand, the essential TCA cycle protein DLAT undergoes lipoylation mediated by FDX1, and the lipoylated protein's binding with copper triggers copper toxicity [11]. Additionally, a recent study demonstrated that ferroptosis inducers sorafenib and erastin not only induce ferroptosis but also promote cuproptosis. The underlying mechanism involves inhibiting the degradation of FDX1 mediated by mitochondrial matrix-associated proteases and suppressing intracellular GSH synthesis [19]. This suggests that intracellular GSH synthesis may act as a common mediator between ferroptosis and cuproptosis [17].

The resistance of tumor cells to chemotherapy has long been a significant challenge in cancer treatment. Modulating cuproptosis and ferroptosis holds promise as a novel approach to overcoming tumor resistance to therapy [20]. Moreover, the interplay between cuproptosis, ferroptosis, and the tumor immune microenvironment has emerged as a critical factor in addressing resistance to tumor immunotherapy [17]. Previous studies have demonstrated that cuproptosis- and ferroptosis-related subtypes in tumors such as hepatocellular carcinoma and ovarian cancer exhibit distinct immune microenvironment characteristics and prognostic differences [21, 22]. However, whether similar patterns exist in colon cancer remains unclear. Therefore, constructing molecular subtypes associated with both cuproptosis and ferroptosis and exploring their corresponding immune microenvironment features are essential for predicting prognosis and guiding therapeutic decision-making.

## Materials & methods

### Collection of data and genes

RNA-seq data and clinical information of colon cancer patients were obtained from The Cancer Genome Atlas (TCGA) (https://portal.gdc.cancer.gov/) and the Gene Expression Omnibus (GEO) database (https://www.ncbi.nlm.nih.gov/gds/). The TCGA dataset was partitioned into training and validation sets, with a predictive model constructed using the training set and validated on the validation set. External validation was further performed on the GSE39582 dataset from the GEO database. Additionally, a total of 488 ferroptosis-related genes and 27 cuproptosis-related genes were obtained from FerrDb (http://www.zhounan.org/ferrdb/) for subsequent bioinformatics analysis [23], as presented in S1 Table.

### Analysis of colon cancer molecular subtypes defined by FCDEGs

In order to identify ferroptosis and cuproptosis related differential expression genes (FCDEGs), we utilized the "limma" R package to perform differential analysis between normal and tumor samples in the TCGA cohort, with criteria set as |log2FC| > 1 and FDR < 0.05. Subsequently, we constructed protein-protein interaction (PPI) networks for differentially expressed genes related to ferroptosis and cuproptosis using the String database (https://string-db.org/). Then, based on the expression profiles of differentially expressed genes related to ferroptosis and cuproptosis, we performed unsupervised consensus clustering of colon cancer samples from TCGA using the "ConsensusClusterPlus" R package to identify distinct subtypes. The ESTIMATE algorithm was employed to analyze differences in tumor microenvironment (TME) scores among different molecular subtypes, and differences in ssGSEA scores between subtypes were compared to assess immune cell infiltration and immune functional differences.

### Construction and validation of prediction model

Our research utilized the "limma" R package to perform differential analysis between ferroptosis and cuproptosis-related subtypes within the TCGA cohort, identifying differentially expressed genes between these subtypes. Criteria for selection included fold change (FC) > 1.5, |log2FC| > 0.584963, and false discovery rate (FDR) < 0.05. Subsequently, univariate COX analysis was conducted to identify genes associated with prognosis (P < 0.05). Samples with a survival time of 0 were excluded, and the TCGA cohort was divided into training and testing sets in a 5:5 ratio using the "caret" R package. For the training set, LASSO Cox regression analysis was performed using the "glmnet" R package to select optimal prognostic genes. Multivariable Cox regression analysis was then carried out using the "survival" and

"survminer" R packages, and a stepwise regression method based on the Akaike information criterion (AIC) was employed to select genes for constructing the prognostic model. KM curves were used to perform prognostic analysis of the model-related genes. The risk score for each patient was calculated using the following formula: Risk Score = $\sum$ (expi * coefi), where exp and coef represent the expression level and corresponding coefficient of the respective gene. Patients in the training set were stratified into high-risk (> median) and low-risk (< median) groups based on their risk scores.

We employed the "survival" R package to generate Kaplan-Meier survival curves for analyzing the survival rates between high-risk and low-risk groups. Concurrently, the "timeROC" R package was utilized to construct receiver operating characteristic (ROC) curves, validating the specificity and sensitivity of the prognostic model. Subsequently, utilizing the aforementioned formula, we obtained the risk scores for each patient in the test set and the GEO database GSE39582 cohort. Using the same approach, Kaplan-Meier survival curves and ROC curves were plotted to assess the survival differences between high-risk and low-risk groups and to validate the effectiveness of the prognostic model.

## Clinical information analysis

To assess the clinical significance of the model, we compared the risk score differences across different ages, genders, tumor grades, and stages between TCGA and GEO cohort samples. We utilized the "ggpubr" R package to draw boxplots. To enhance the predictive accuracy and clinical utility of the model, we employed the "rms" and "regplot" R packages to construct a nomogram prediction model based on clinical information and risk scores. Calibration curves were plotted to evaluate the accuracy of the model.

## Immune infiltration and immunotyping analysis

We utilized the single-sample gene set enrichment analysis (ssGSEA) function within the "GSVA" R package to compute the levels of 28 immune cell types and 13 immune functions for each sample in the TCGA and GEO cohorts. The immune infiltration differences between the high-risk and low-risk groups were compared using the Wilcoxon test. The 28 immune cell types and 13 immune functions involved in this study were integrated from previous relevant research [24–26], and the immune feature gene sets are presented in S2 Table. Spearman correlation analysis was conducted to examine the correlation between the seven genes in the predictive model and the risk score with the 28 immune cell types and 13 immune functions in the TCGA cohort. A correlation heatmap was generated using the "pheatmap" R package. Subsequently, we further explored the relationship between the seven model genes and the risk score with the immune infiltration microenvironment in the TCGA cohort. Molecular immune subtyping data from previous studies [27] were obtained for the TCGA cohort to compare the differences in risk scores between different immune subtypes.

## Functional enrichment analysis

Using the "clusterProfiler" R package to conduct gene ontology and Kyoto Encyclopedia of Genes and Genomes enrichment analysis on differential genes between ferroptosis and cuproptosis related subtypes, and visualizing the results using the "enrichplot" R package.

## Gene set variation analysis

We downloaded Kyoto Encyclopedia of Genes and Genomes pathway gene set data from the MSigDB database (http://www.gsea-msigdb.org/), and employed the "GSVA" R package to

compute gene set variation enrichment scores for each sample in the TCGA cohort. Subsequently, we utilized the "limma" R package to compare the differences in enrichment scores between high-risk and low-risk groups, and visualized the results using the "pheatmap" R package. The filtering criteria were set as |log2FC| > 0.15 and FDR < 0.05.

## Drug sensitivity analysis

Our research obtained the cell line expression matrix and corresponding IC50 values for drugs from the cancer therapy response portal (http://portals.broadinstitute.org/ctrp.v2.1/), which is hosted on the website developed by the authors of the "oncoPredict" R package (https://osf.io/c6tfx/). This dataset was compiled and utilized as the training set. Using the "oncoPredict" R package, we calculated drug sensitivity scores for each sample in the TCGA cohort. Subsequently, we conducted drug sensitivity score-based screening to identify drugs sensitive to colon cancer. Furthermore, we performed correlation analysis between drug sensitivity scores, risk scores, and the expression levels of seven genes within the model.

## Real-time quantitative PCR (qRT-PCR)

This study collected 10 pairs of fresh colon cancer tissues and adjacent non-cancerous tissues from colon cancer patients at the First Affiliated Hospital of Chongqing Medical University between March 1, 2024, and March 30, 2024. The study has been approved by the Ethics Committee of the First Affiliated Hospital of Chongqing Medical University (approval number K2024-058-01). Total RNA was extracted from the cancer tissues and adjacent non-cancerous tissues using TRIzol reagent, followed by reverse transcription of total RNA into total cDNA using Evo M-MLV RT Reaction Mix Ver.2. Subsequently, cDNA was amplified using designed primers with the following sequences: ASRGL1 (forward 5'-CAGGTTGTGGGTCTGTCTTGAA-3', reverse 5'-GCCGAGCAAGTTTAATGGGATTT-3'), EREG (forward 5'-ATCACAGTCGTCGGTTCCACATA-3', reverse 5'-TGACCTAACACTTGACCCAACAT-3'), RORC (forward 5'-GAGCATGGAGGAGGAAAGTTTGA−3', reverse 5'-ATTTGTGAGGTGTGGGTCTTCTT-3'), C12orf56 (forward 5'-TACTTGCCGGAGTCTAGGGATAA-3', reverse 5'-CGTGATGACTCGGTTTCTGTTTC-3'), MAGEA12 (forward 5'-ACAGAGGGCCCCCAATAATC-3', reverse 5'-GTGTTGACCTGAGTCACCCT-3'), MMP10 (forward 5'-GGAAGCTAGACACTGACACTCTG-3', reverse 5'-TCAACAGCATCTCTTGGCAAATC-3'), INHBB (forward 5'-AAATCATCAGCTTCGCCGAGAC−3', reverse 5'-GTAGGGCAGGAGTTTCAGGTAAA-3'), GAPDH (forward 5'-TGTCAAGCTCATTTCCTGGTATG-3', reverse 5'-TCTCTCTTCCTCTTGTGCTCTTG-3'). Gene transcription levels were calculated using the $2^{-\Delta\Delta Ct}$ method with GAPDH as the reference. Finally, a paired t-test was used to validate the differential expression of the seven model-related genes between colon cancer tissues and adjacent non-cancerous tissues.

## Immunohistochemistry

We further validated the differential expression of model-related genes between tumor tissues and adjacent non-cancerous tissues using immunohistochemistry. First, 70 pairs of colon cancer and adjacent tissue sections were deparaffinized in xylene and rehydrated in a series of ethanol solutions of varying concentrations, followed by antigen retrieval and removal of endogenous peroxidase. The anti-EREG (YT6045, 1:200) and anti-MMP10 (YT2793, 1:300) antibodies from *Immunoway*, the anti-INHBB (FNab04320, 1:100) and anti-ASRGL1 (FNab00648, 1:50) antibodies from *FineTest*, the anti-MAGEA12 (bs-6818R, 1:200) antibody from *Bioss*, the anti-RORC (HA722121, 1:500) antibody from *HUABIO*, and the anti-C12orf56 (HPA056986, 1:1000) antibody from *Atlas Antibodies* were each applied to 10 pairs of tissue slides and incubated at room temperature for 60 minutes. Polymer HRP (Rabbit)-P

(RS0063), also from *Immunoway*, was applied to the sections and incubated at room temperature for 30 minutes. DAB staining and hematoxylin counterstaining were then performed. Two pathologists, blinded to the sample information, evaluated the immunohistochemistry results as follows: cells with <10% staining was scored as 1; cells with 10–49% staining as 2; cells with 50–74% staining as 3; and cells with 75–100% staining as 4. The intensity of staining was scored as 1 for light yellow granules, 2 for yellow-brown granules, and 3 for brown granules. The final score (IRS Score) was determined by multiplying the percentage of stained cells by the intensity score [28]. A paired t-test was used to validate the expression differences of the gene between colon cancer tissues and adjacent non-cancerous tissues.

## Results

### Differentially expressed genes related to ferroptosis and cuproptosis in colon cancer

The specific workflow of this study is shown in Fig 1. We obtained a total of 524 cases, consisting of colon cancer tumor samples and matched normal samples, from TCGA. For each patient, only one tumor sample and one normal sample were retained. Samples deemed unsuitable for analysis were filtered out based on the annotation file in the TCGA database. Ultimately, we selected 37 normal samples and 438 colon cancer tumor samples, and the excluded samples are listed in S3 Table. A total of 509 ferroptosis and cuproptosis-related genes were collected from the FerrDb website, excluding the two genes RNF113A and TAFAZZIN, which were not present in the TCGA cohort. Differential gene expression analysis between colon cancer tumor samples and normal samples in the TCGA cohort identified 103 ferroptosis and cuproptosis related differential expression genes (FCDEGs) (Fig 2A and 2B). Protein-protein interaction (PPI) network analysis revealed interactions among these genes, with genes ranked by betweenness centrality. The top 10 genes were considered as key nodes in the network (Fig 2C).

### The subtypes associated with ferroptosis and cuproptosis

Consistency clustering analysis was conducted on 103 ferroptosis and cuproptosis-related differential expression genes (FCDEGs). Based on the matrix heatmap, cumulative distribution function (CDF) plot, and relative change in area under the CDF curve, K = 2 was determined to be the optimal clustering, indicating maximum stability (Fig 3A–3C). Kaplan-Meier survival analysis suggested no difference in survival time between the two subtypes (S1 Fig). Subsequently, we employed ssGSEA to analyze immune cell infiltration and immune function between the two subtypes, revealing differences in some immune cell infiltrations and immune functions between the subtypes (Fig 3D and 3E), with significant differences observed in Th17 and Th2 cells. The TME scores were calculated for each sample, and differences in TME scores between the two subtypes were compared, revealing differences in stromal scores in tumor tissues, while there were no differences in immune cell infiltration scores and tumor purity scores in tumor tissues (Fig 3F).

### The construction and validation of the model

Differential analysis between the two subtypes yielded ferroptosis and cuproptosis subtypes related differential expression genes (FCSDEGs). Subsequently, univariate Cox regression analysis was performed on these differential genes, resulting in 52 prognosis-related genes (S4 Table). Next, the TCGA cohort was divided into training set and testing set in a 5:5 ratio. LASSO Cox regression was conducted on the training set for the 52 prognosis-related genes to

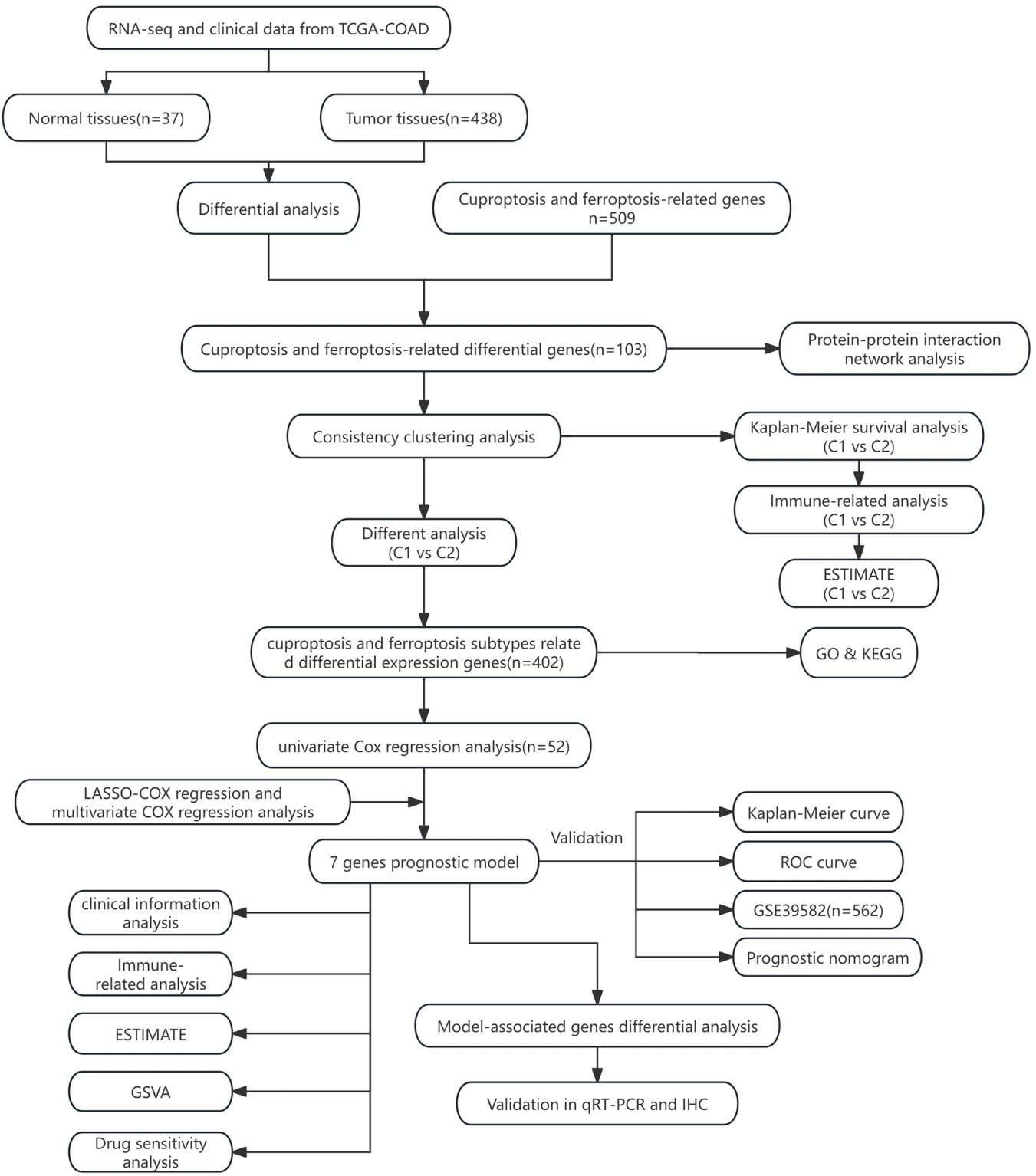

**Fig 1. The workflow of the study.**

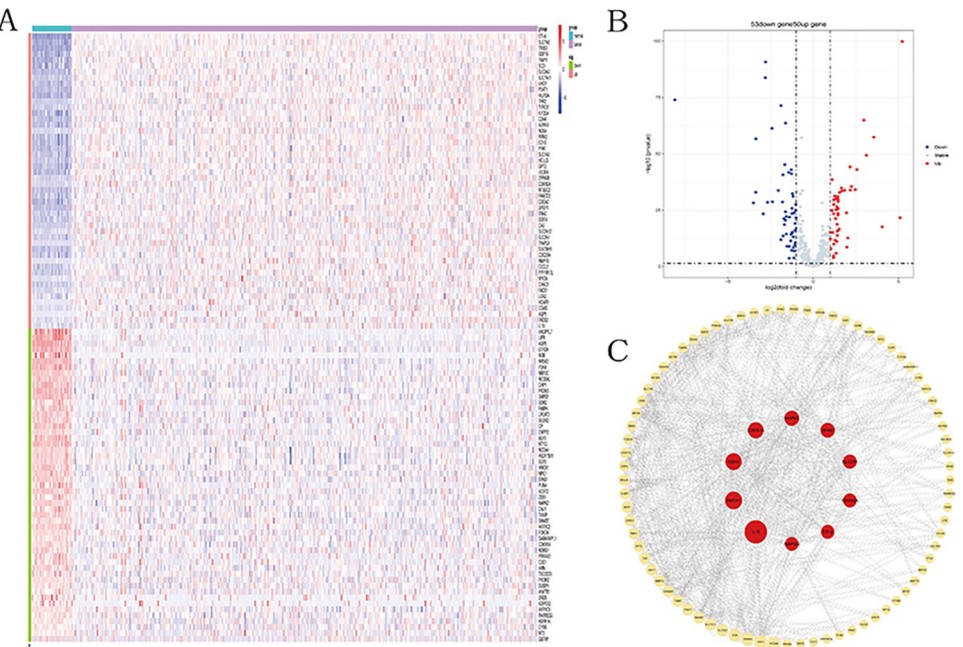

**Fig 2. Screening and analysis of cuproptosis and ferroptosis-related differential genes.** (A) Heatmap of CFDEGs in tumor and normal groups. Blue: low expression level; Red: high expression level. (B) Volcano plot of cuproptosis and ferroptosis-related genes. (C) Protein-protein interaction network among CFDEGs (cutoff = 0.4). Core genes among CFDEGs.

identify important prognostic genes (Fig 4A and 4B). Finally, a prognostic model of 7 genes (MAGEA12, C12orf56, ASRGL1, EREG, RORC, MMP10, INHBB) was formed through multi-variate COX regression analysis (Fig 4C). Kaplan-Meier survival analysis showed a significant difference in survival time between the high and low expression groups of MAGEA12, ASRGL1, and INHBB (S2 Fig). The risk score formula: Risk score = 0.134740737082723 * MAGEA12 + 0.327294645435294 * C12orf56−0.290542928969321 * ASRGL1− 0.188227916852889 * EREG—0.178355841003148 * RORC—0.160071259389688 * MMP10 + 0.112345147800628 * INHBB. Using the median risk score of patients in the training set of the TCGA cohort as the cutoff, patients were divided into high-risk and low-risk groups (Fig 4D). Scatter plot analysis revealed a higher mortality rate in the high-risk group compared to the low-risk group (Fig 4E). Heatmap analysis indicated higher expression levels of ASRGL1, EREG, RORC, and MMP10, and lower expression levels of MAGEA12, C12orf56, and INHBB in the high-risk group, consistent with the model conclusion (Fig 4F). Kaplan-Meier survival analysis comparing survival differences between the two groups showed that patients in the high-risk group had shorter survival times than those in the low-risk group (Fig 5A). ROC curves of the training set indicated an AUC of 0.747, 0.736, and 0.775 at 1 year, 3 years, and 5 years respectively, confirming the reliability of the model (Fig 5D).

In both the testing set of the TCGA cohort and the GEO cohort, we computed the risk scores for each sample using the aforementioned formula. Scatter plots and heatmaps were generated using the same method as the training set. Consistently, patients in the high-risk group exhibited higher mortality rates compared to the low-risk group. However, the heatmap of the GEO cohort indicated no significant difference in the expression level of RORC between the high-risk and low-risk groups (Fig 4G–4L). Kaplan-Meier survival curves demonstrated that the survival time of high-risk patients in both the testing set of the TCGA cohort and the

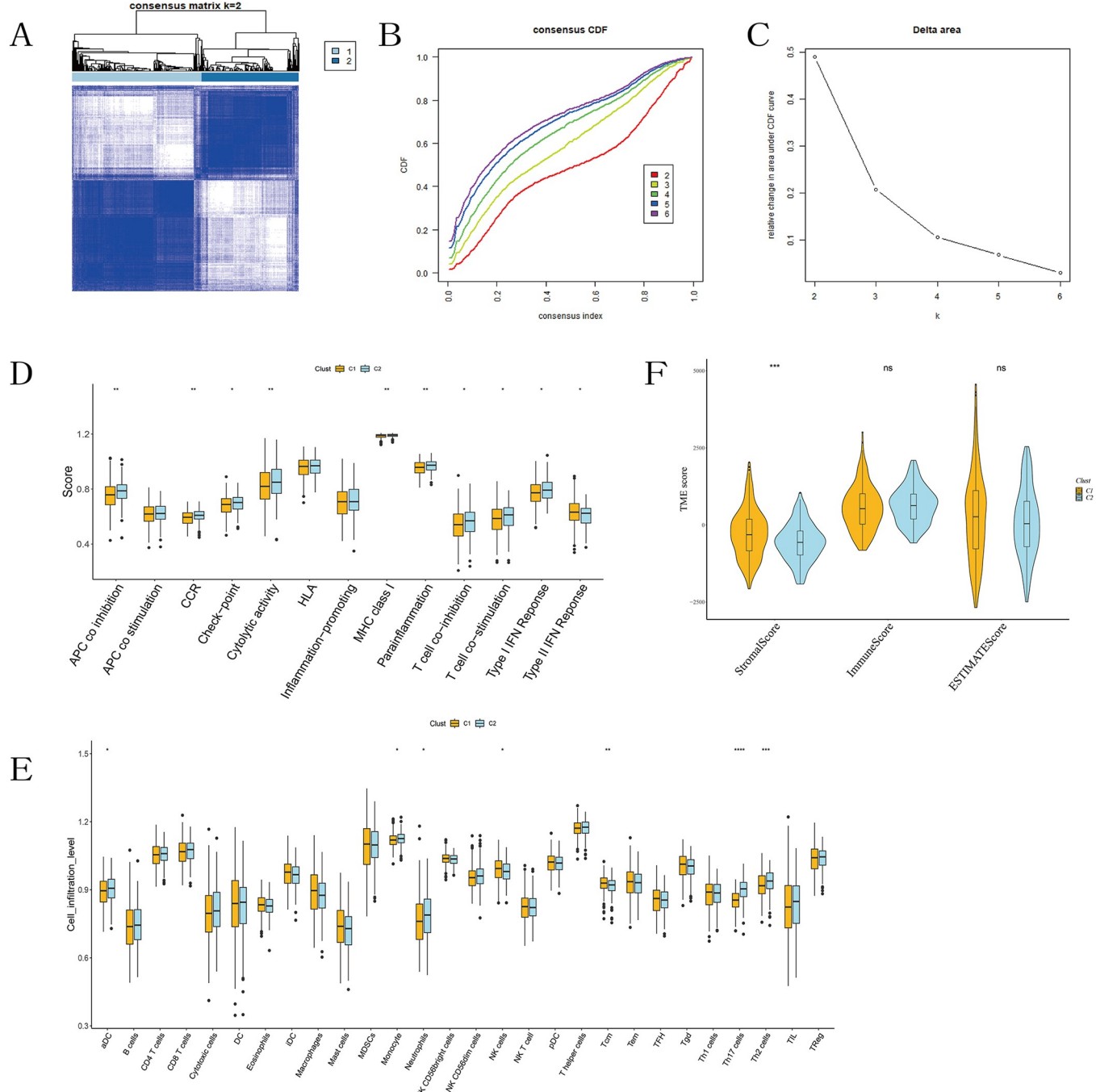

**Fig 3. Identification of subtypes related to cuproptosis and ferroptosis-related differential genes.** (A) Two different gene clusters (k = 2). (B) CDF of consensus clustering K from 2 to 6. (C) Relative change in the area under the CDF curve (k from 2 to 6). (D) Scores of 13 immune functions between the two subtypes. (E) Scores of 28 immune cell infiltrations between the two subtypes. (F) Differences in immune, stromal, and estimate scores between the two subtypes.

GEO cohort was lower than that of the low-risk group (Figs 5B and 4C). The ROC curves of both cohorts confirmed the applicability of the predictive model in other datasets as well (Fig 5E and 5F).

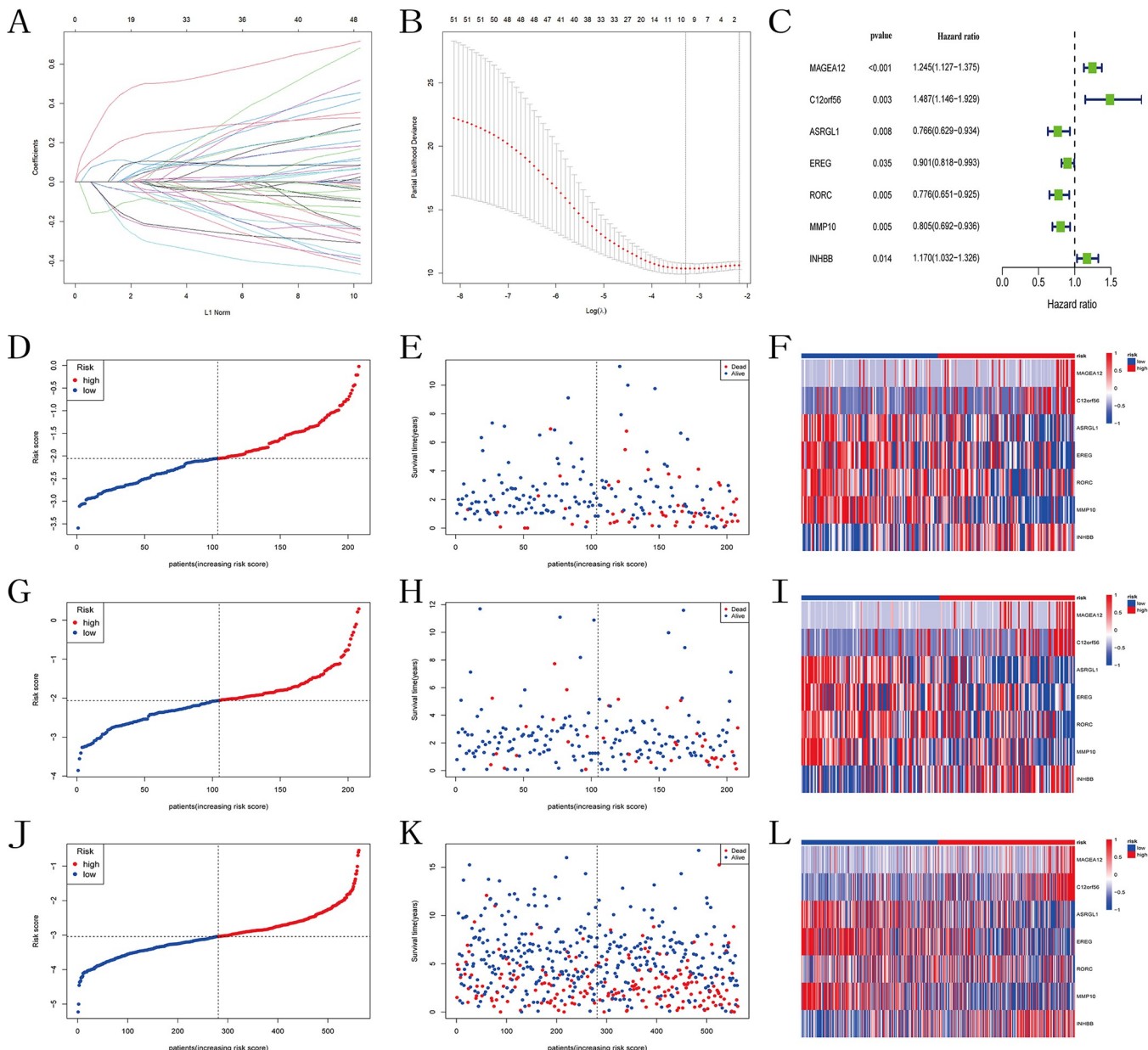

**Fig 4. Construction of the prognostic model.** (A) LASSO coefficient plot of prognostic genes in the TCGA training set. (B) Cross-validation for LASSO regression coefficient selection. (C) Univariate COX regression analysis of the selected prognostic genes in the TCGA training set. (D-E) Risk score and survival status distribution plots in the TCGA training set, (G-H) TCGA testing set, (J-K) GEO cohort. (F) Heatmap of model-related gene expression in the TCGA training set, (I) TCGA testing set, (L) GEO cohort.

## The clinical significance of the prognostic model

We analyzed the relationship between clinical information from TCGA and GEO cohorts and the predictive model to assess the clinical significance of the model. In the TCGA cohort, we found that the risk score was not associated with patient age or tumor location; however, female patients had higher risk scores compared to male patients, and higher tumor grades were associated with higher risk scores. Similar conclusions were drawn from the GEO cohort regarding patient age, tumor location, and tumor grade. However, unlike the TCGA cohort,

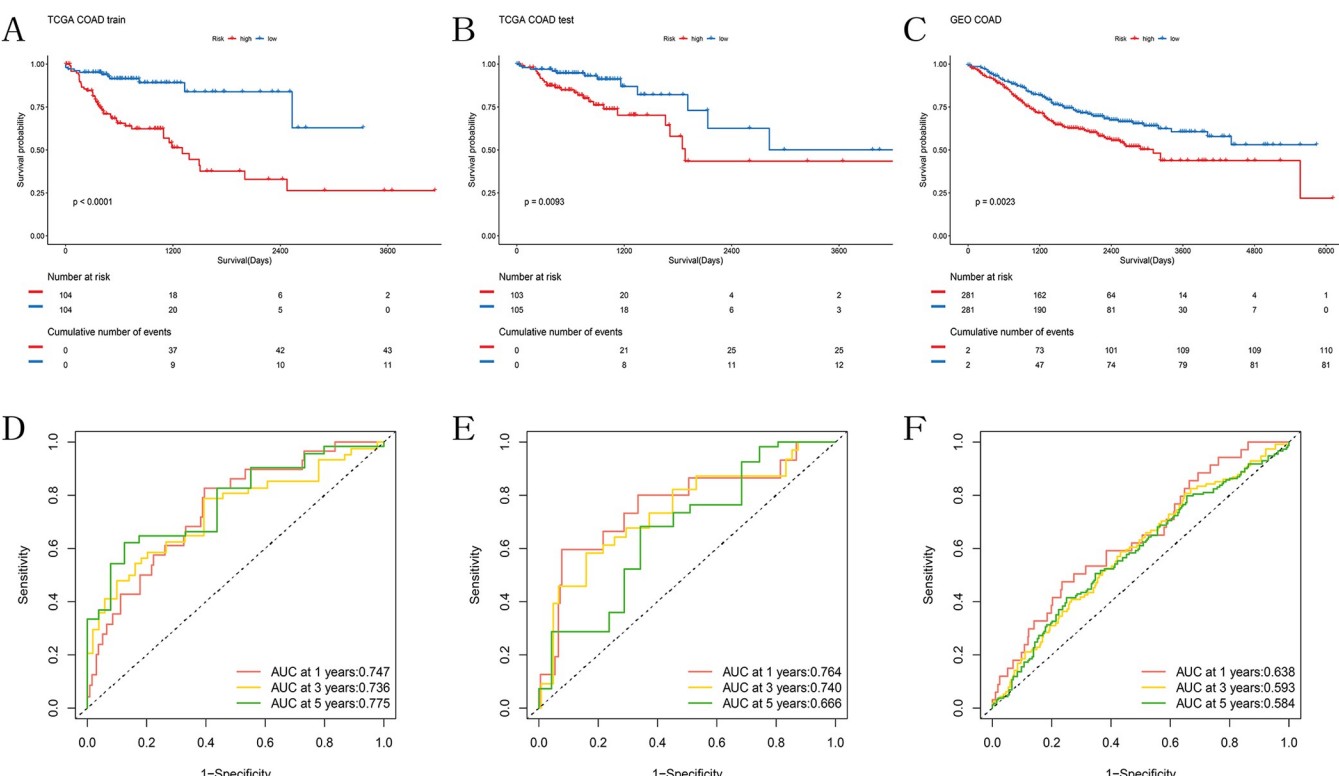

**Fig 5. Validation of the prognostic model.** (A) KM curves of OS for two risk groups in the TCGA training set, (B) TCGA testing set, (C) GEO cohort. (D) ROC curves for OS over time in the TCGA training set, (E) TCGA testing set, (F) GEO cohort.

we found no significant relationship between risk scores and patient gender in the GEO cohort (Fig 6A–6H).

## Plotting and analysis of nomogram

We further analyzed whether clinical features from the TCGA and GEO cohorts could serve as independent prognostic indicators through univariate Cox regression and multivariate Cox regression. The results indicated that age, tumor grade, and risk score could act as independent prognostic variables for colon cancer in both the TCGA and GEO cohorts (Fig 7A–7D). Subsequently, we generated nomograms based on the independent prognostic indicators identified in the TCGA cohort (Fig 7E) and calculated the nomogram score for each sample. The ROC curves demonstrated AUC values of 0.796, 0.802, and 0.796 for the model's predictive ability at 1, 3, and 5 years of survival, respectively (Fig 7F). Further calibration curves illustrated that the model provided accurate predictions of survival (Fig 7G).

## Functional enrichment analysis and gene set variation analysis

Functional enrichment analysis was conducted on the ferroptosis and cuproptosis subtypes related differential expression genes (FCSDEGs) to explore the biological functional differences between the two subtypes. GO analysis revealed significant enrichment of differential genes in biological processes such as extracellular matrix organization, extracellular structure organization, external encapsulating structure organization, collagen-containing extracellular matrix, signal receptor activator activity, and receptor ligand activity (Fig 8A). KEGG analysis results indicated significant enrichment of differential genes in processes such as cytokine-

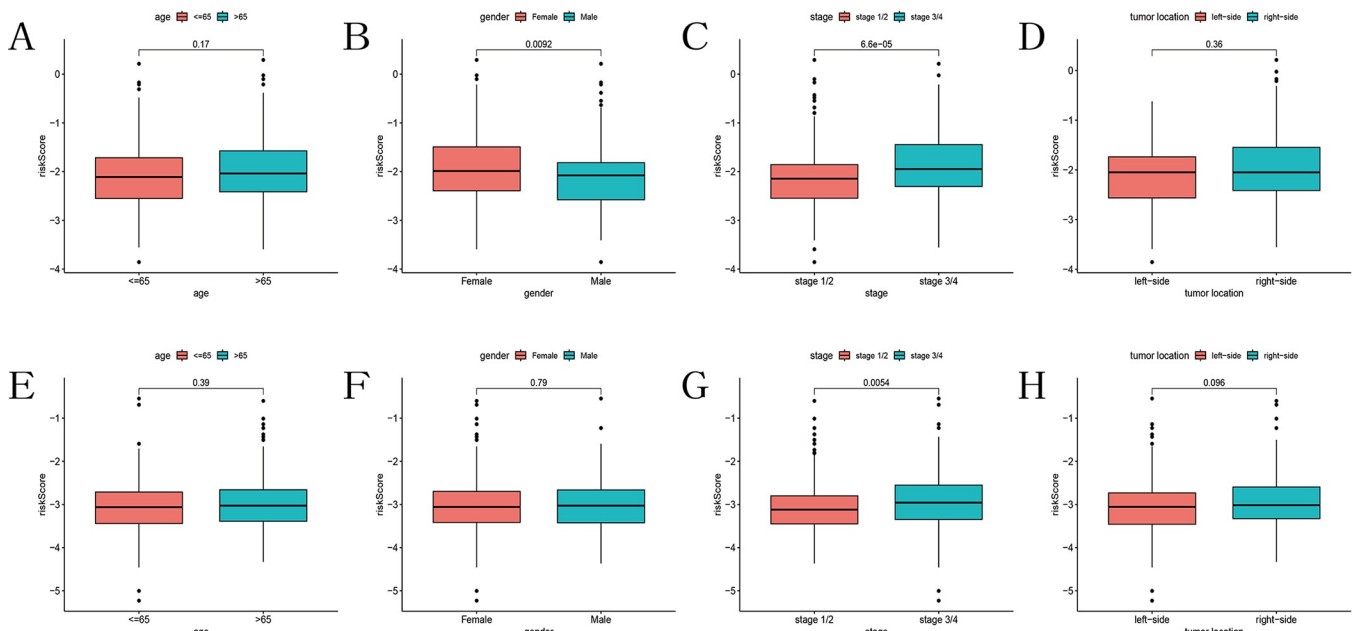

**Fig 6. Differences in risk scores among various clinical features.** Differences in risk scores among (A) age, (B) gender, (C) tumor stage, (D) tumor location in the TCGA cohort, and among (E) age, (F) gender, (G) tumor stage, (H) tumor location in the GEO cohort.

cytokine receptor interaction and the IL-17 signaling pathway (Fig 8B). Furthermore, we performed gene set variation analysis on KEGG canonical pathway gene sets and compared the differences between the high-risk and low-risk groups. When the cutoff was set at |log2FC| > 0.15 and FDR < 0.05, we found differences between the two groups in 24 pathways, including the Nod-like receptor (NLR) signaling pathway, fructose and mannose metabolism, and apoptosis (Fig 8C).

## Immune characteristics and Tumor Microenvironment (TME) analysis

In this study, we utilized single-sample gene set enrichment analysis (ssGSEA) to assess the enrichment levels of immune cells and immune functions in each sample of the TCGA cohort. We found that in the low-risk group, activated dendritic cells (aDC), CD4 T cells, neutrophils, Th17 cells, Th2 cells, and 19 other types of immune cells were significantly enriched (Fig 9A). Conversely, in the high-risk group, nine immune functions, including antigen-presenting cell inhibition, CC chemokine receptor signaling, checkpoint regulation, and para-inflammation, were observed to be decreased (Fig 9B). Furthermore, we generated correlation heatmaps depicting the associations between seven model genes, risk scores, and immune cells as well as immune functions. The immune cell heatmap revealed positive correlations of ASRGL1 with Th17 cells (r = 0.39, p<0.01) and Th2 cells (r = 0.44, p<0.01), MMP10 with neutrophils (r = 0.40, p<0.01), activated dendritic cells (r = 0.30, p<0.01), dendritic cells (r = 0.36, p<0.01), and Th17 cells (r = 0.36, p<0.01). Conversely, the risk score of the model was negatively correlated with Th17 cells (r = -0.38, p<0.01) (Fig 9C). The immune function heatmap revealed a positive correlation between MMP10 and CC chemokine receptor signaling (r = 0.41, p<0.01), while no significant correlations were observed between other genes and immune characteristics (Fig 9D).

Subsequently, we further analyzed the correlation between ASRGL1 and gene sets related to Th17 cells and Th2 cells, as well as the correlation between MMP10 and gene sets related to

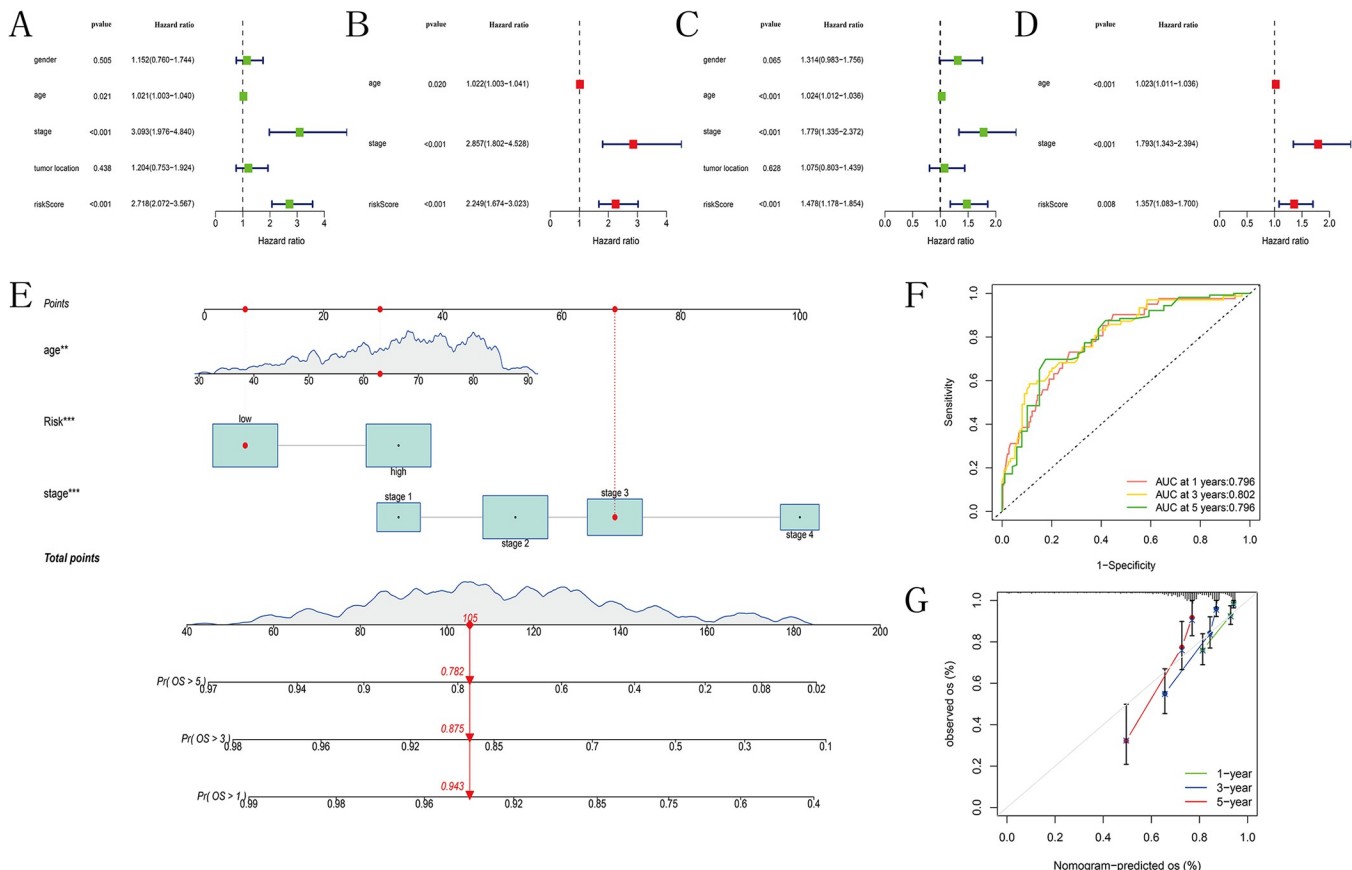

**Fig 7. Establishment and evaluation of the nomogram.** (A-B) Univariate and multivariate COX regression analysis of risk scores combined with clinical features and OS in the TCGA cohort. (C-D) Univariate and multivariate COX regression analysis of risk scores combined with clinical features and OS in the GEO cohort. (E) Nomogram predicting 1-year, 3-year, and 5-year OS for colon cancer patients. (F) ROC curve of the nomogram. (G) Calibration curves of the nomogram's predicted versus actual survival.

neutrophils, activated dendritic cells (aDC), dendritic cells (DC), and Th17 cells within the TCGA cohort. The results revealed that ASRGL1 exhibited correlation with C2CD4A in the Th17 cell gene set and with MB, RAB27B, PDE4B, and RNF125 in the Th2 cell gene set (Fig 9E). MMP10 demonstrated correlation with SLC25A37 in the neutrophil gene set, with CCL22 in the dendritic cell gene set, and with NOS2 in the activated dendritic cell gene set (Fig 9F).

Furthermore, we evaluated the tumor microenvironment in the TCGA cohort using the ESTIMATE algorithm. We found no significant difference in the stromal cell score between the high-risk and low-risk groups, while the immune cell score was higher in the low-risk group compared to the high-risk group, indicating higher immune infiltration in the low-risk group, consistent with the above results. ESTIMATE scores indicated no difference in tumor purity between the two groups (Fig 9G). We also compared the risk scores between the four molecular immune subtypes, including wound healing (C1), interferon-gamma (IFN-γ) dominant (C2), inflammatory (C3), and lymphocyte depleted (C4). We observed that the risk score of the inflammatory subtype (C3) was significantly higher than that of the wound healing (C1) and interferon-gamma (IFN-γ) dominant (C2) subtypes (Fig 9H).

## Drug sensitivity analysis

Utilizing the "oncoPredict" R package, we computed sensitivity scores for each sample in the TCGA cohort across 198 drugs, and selected drugs with an average sensitivity score below 5

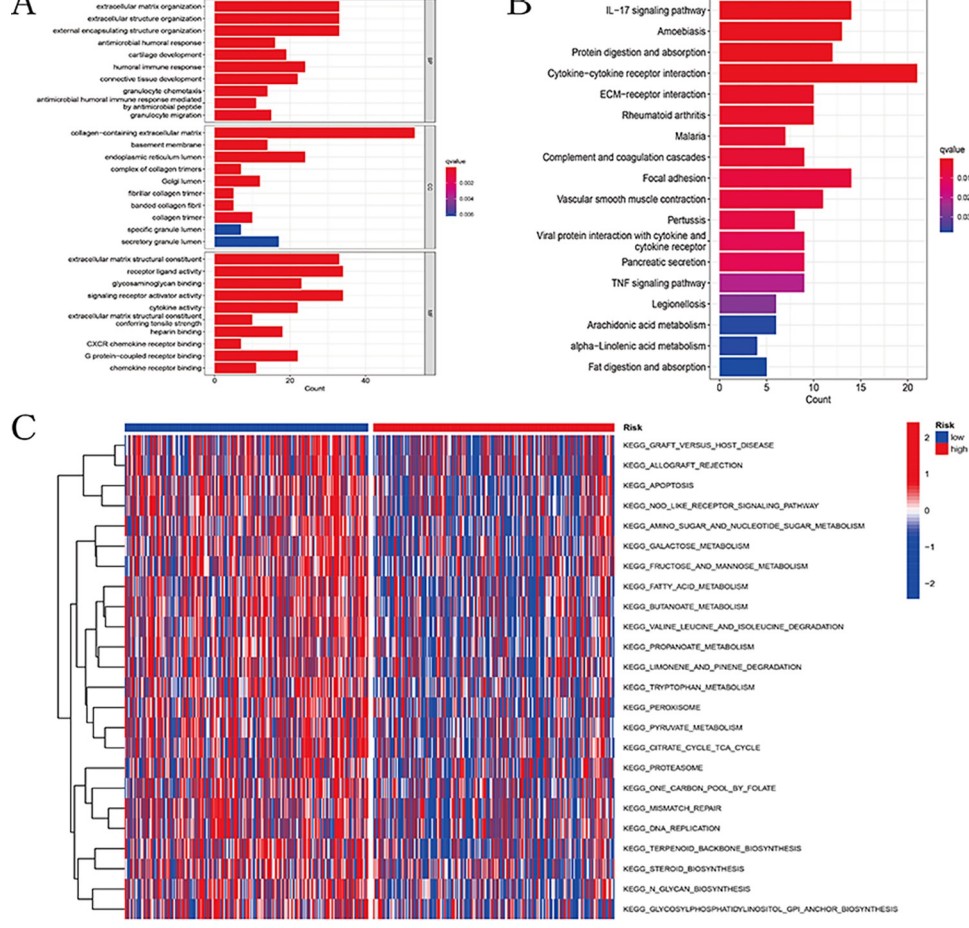

**Fig 8. Enrichment analysis of differential genes between cuproptosis and ferroptosis-related subtypes in the TCGA cohort.** (A) Bar plot of GO enrichment analysis. (B) Bar plot of KEGG pathways. (C) Heatmap of GSVA analysis between high and low-risk groups in the TCGA cohort.

for subsequent investigation. Subsequently, we further analyzed the correlation between the sensitivity scores of selected drugs and seven model genes (cor>0.3, p<0.05). We generated scatter plots for the top 6 drugs for each gene based on their correlation coefficients. The results revealed that ASRGL1 was primarily associated with sensitivity to Dactinomycin_1911, Docetaxel_1007, Eg5_9814_1712, Epirubicin_1511, Pevonedistat_1529, and Buparlisib_1873 (Fig 10A–10F). EREG showed significant correlation with Dactinomycin_1911, Epirubicin_1511, Foretinib_2040, Mitoxantrone_1810, Sabutoclax_1849, and Eg5_9814_1712 (Fig 10G–10L). Other genes showed no correlation with drug sensitivity scores. Furthermore, we analyzed the differences in drug sensitivity scores between the high-risk and low-risk groups and found that sensitivity to AZD8055_1059, Bortezomib_1191, Dihydrorotenone_1827, and MG-132_1862 was higher in the low-risk group compared to the high-risk group (Fig 10M–10P).

## Model-associated genes differential analysis

We analyzed the expression differences of seven model-associated genes between normal and tumor samples in the TCGA database. The results showed that, in the TCGA cohort, C12orf56 and RORC had higher expression in normal samples, while EREG, INHBB, MAGEA12, and

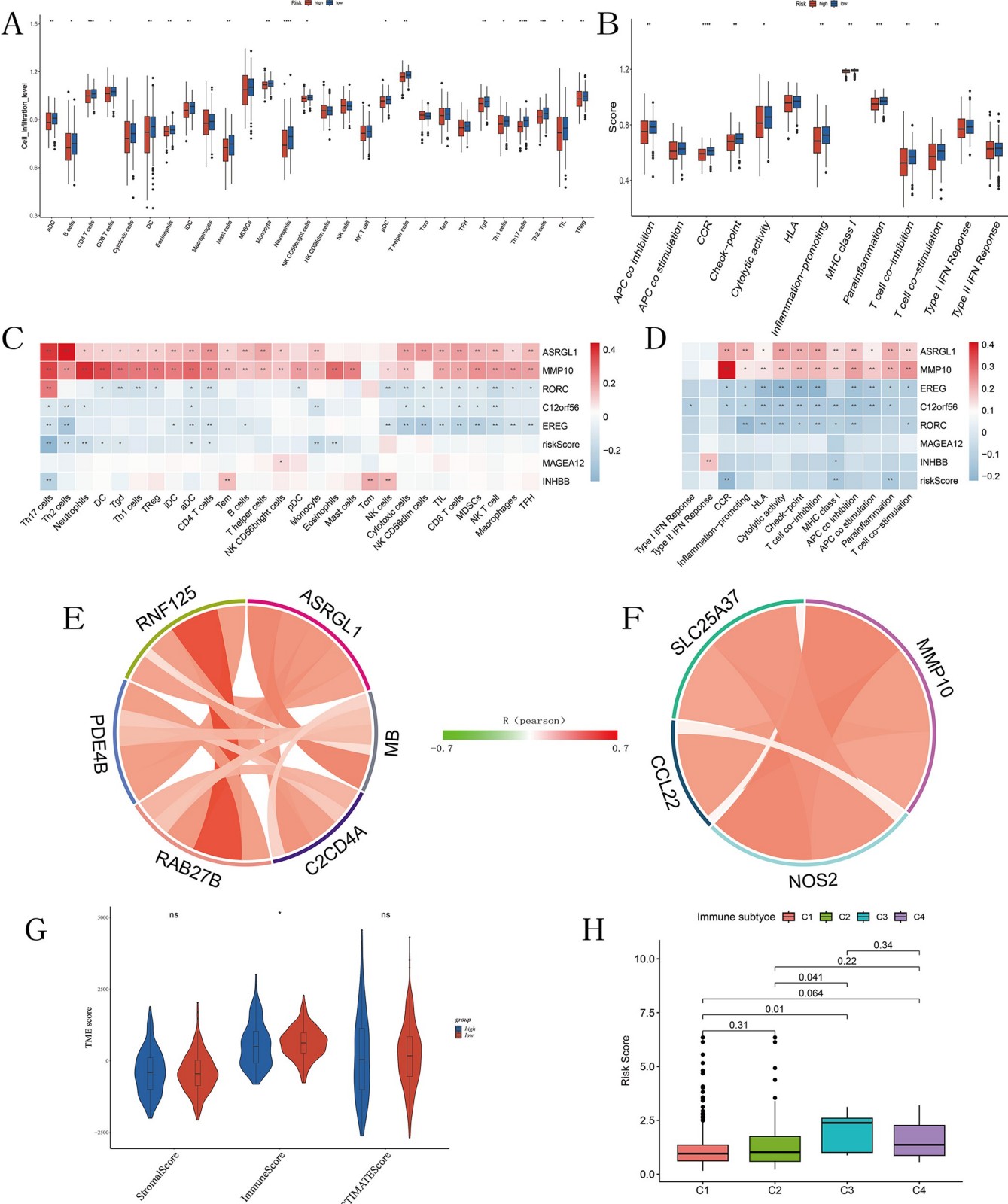

**Fig 9. Analysis of immune status and tumor microenvironment in the TCGA cohort.** (A) Scores of 28 immune cell infiltrations between high and low-risk groups. (B) Scores of 13 immune functions between high and low-risk groups. (C) Heatmap of the correlation between risk scores, model-related genes, and 28

immune cells. (D) Heatmap of the correlation between risk scores, model-related genes, and 13 immune functions. (E-F) Correlation of ASRGL1, MMP10 with immune biomarkers in the TCGA cohort. (G) Differences in immune, stromal, and estimate scores between high and low-risk groups. (H) Comparison of risk scores among different immune infiltration subtypes. *P < 0.05, **P < 0.01, ***P < 0.001, ****P < 0.0001.

MMP10 had higher expression in tumor samples; ASRGL1 showed no expression difference between normal and tumor samples (Fig 11A–11G). Subsequently, we validated these findings in the GEO database. Unlike the TCGA results, MAGEA12 showed no expression difference between normal and tumor samples in the GEO dataset, while the analysis results of the other genes were consistent with those of the TCGA database (Fig 11H–11N). Additionally, we

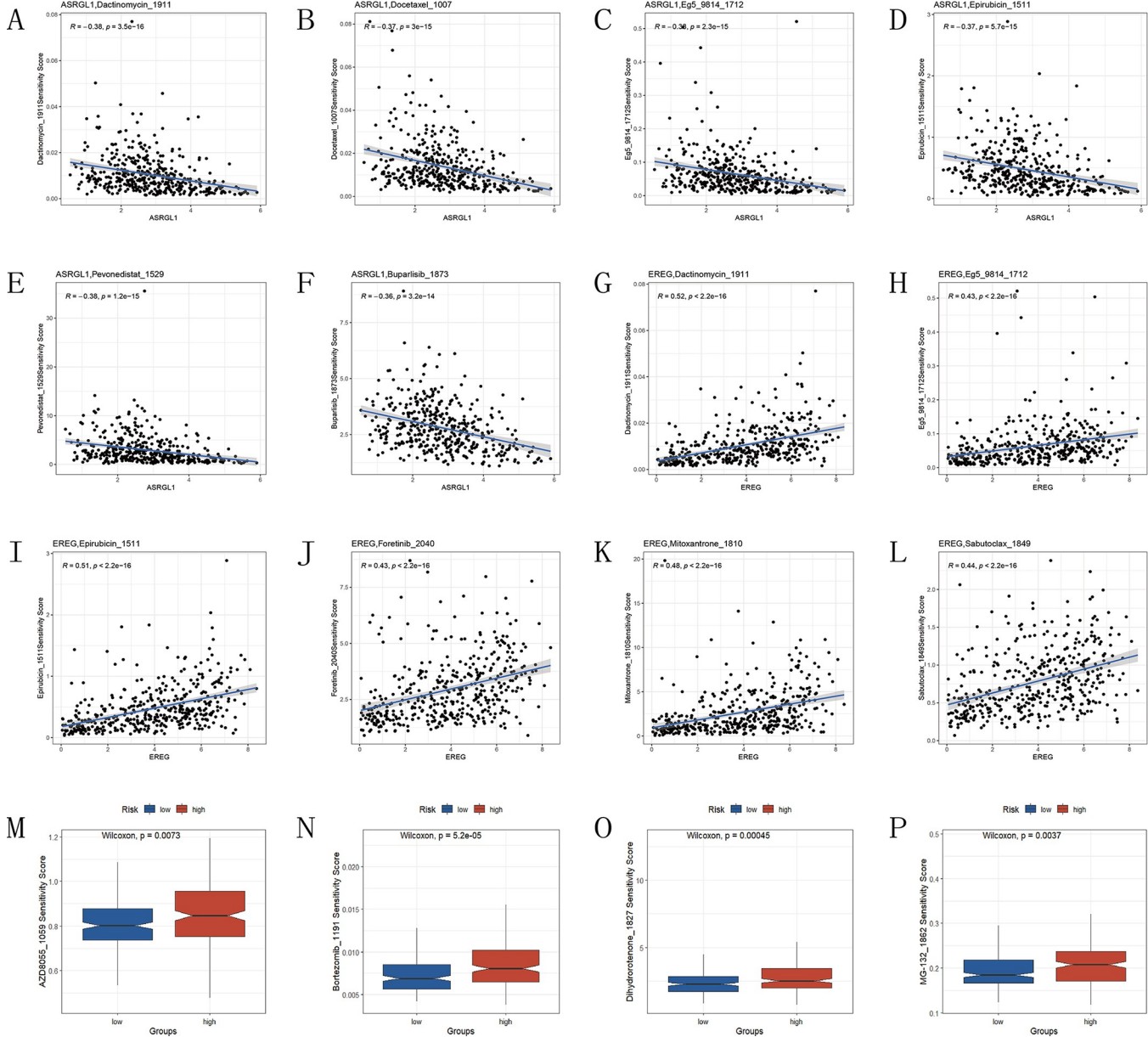

**Fig 10. Drug sensitivity analysis in the TCGA cohort.** (A-F) Scatter plots of drug sensitivity correlation analysis for ASRGL1, (G-L) EREG with the top 6 most correlated drugs (x-axis: gene expression level, y-axis: drug sensitivity score). (M-P) Sensitivity scores of drugs in high and low-risk groups.

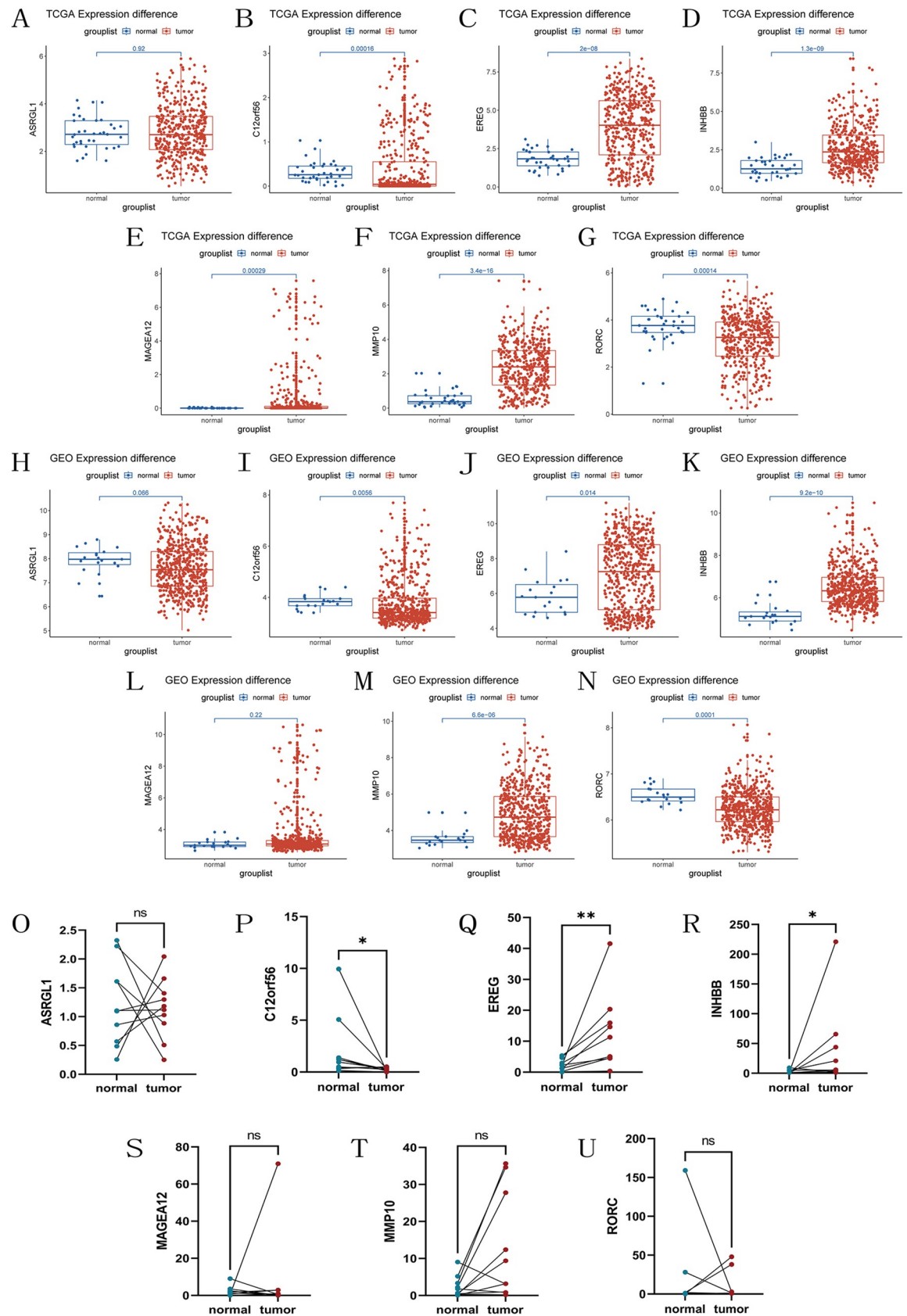

**Fig 11. Differential expression of model-related genes.** Differential expression of 7 model-related genes in colon cancer tumor and normal tissues in the (A-G)TCGA and (H-N)GEO cohorts. Perform qRT-PCR on 10 pairs of fresh colon cancer tissues and adjacent normal tissues obtained from the First Affiliated Hospital of Chongqing Medical University(O-U), *P < 0.05, **P < 0.01, ns = no significant.

verified the expression differences of the seven model-associated genes between colon cancer tissues and adjacent tissues using qRT-PCR. The results showed that there was no difference in the expression of ASRGL1, MAGEA12, MMP10, and RORC between normal and tumor samples. C12orf56 was expressed at higher levels in normal samples, whereas EREG and INHBB were expressed at higher levels in tumor samples (Fig 11O–11U). Furthermore, immunohistochemistry was used to validate the expression differences of model-related genes between colon cancer tissues and adjacent non-cancerous tissues. The results showed that EREG, INHBB, and MMP10 were highly expressed in tumor tissues, C12orf56 and RORC were highly expressed in normal tissues, and the expression of ASRGL1 and MAGEA12 showed no significant difference between tumor and normal tissues (Fig 12).

## Discussion

Colon cancer, ranked as the third most common cancer worldwide, follows only lung cancer in mortality rates. Additionally, the incidence of colon cancer is steadily increasing [29]. A majority of colon cancer deaths occur in low- and middle-income countries, predominantly among patients with advanced-stage tumors. This scenario is closely linked to the dietary patterns prevalent in these nations. Additionally, constrained economic development significantly impedes effective early diagnosis of colon cancer [30, 31]. Despite the current inclusion of various treatment modalities for colon cancer, such as surgery, radiotherapy, targeted therapy, and chemotherapy, neoadjuvant therapy has also become a standard treatment approach. The application of neoadjuvant therapy prior to surgery can reduce tumor volume, lower tumor staging, and enhance the likelihood of completing treatment. However, significant survival benefits have not been consistently observed in previous clinical trials [32, 33]. Therefore, a model for predicting survival in colon cancer patients, particularly those with advanced-stage disease, holds significant value.

Ferroptosis and cuproptosis, two distinct types of cell death, though different, both lead to the demise of tumor cells, thereby inhibiting excessive tumor proliferation. Research has shown that the mitochondrial tricarboxylic acid (TCA) cycle is a convergence point for ferroptosis and cuproptosis [17]. Additionally, copper not only induces protein toxicity stress in cuproptosis but also drives ferroptosis by influencing the degradation of GPX4 [34]. In other words, there is some association between ferroptosis and cuproptosis. Currently, there is no investigation combining ferroptosis and cuproptosis to explore their subtypes and their impact on the survival time of colon cancer patients. This study, utilizing 103 FCDEGs, divided colon cancer patients into two subtypes, revealing the roles of cuproptosis and ferroptosis-related genes in the progression of tumor development. Additionally, the study constructed a colon cancer prognosis model based on differential genes between the two subtypes, further elucidating the biological functional differences between the two subtypes and the immune characteristics of model-related genes.

Our study identified a total of 103 ferroptosis and cuproptosis-related genes that were differentially expressed between colon cancer tissues and normal tissues, indicating the significant roles of ferroptosis and cuproptosis in tumorigenesis and tumor progression. The constructed PPI network revealed that IL1B, HMOX1, CD44, CDKN1A, CDKN2A, MAPK3, EPAS1, SLC2A1, EZH2, and ADIPOQ were among the core genes. CDKN1A and CDKN2A

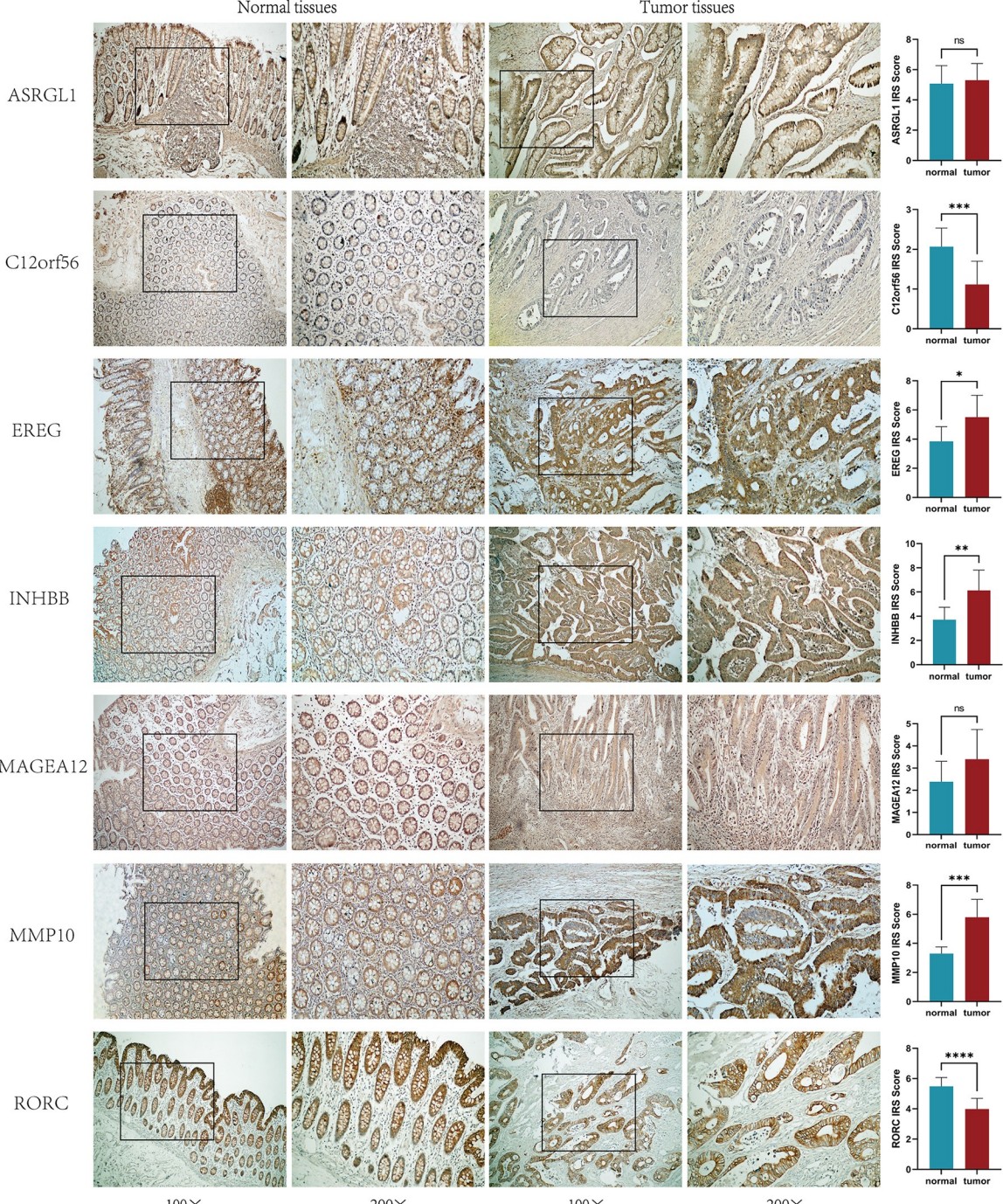

**Fig 12. Immunohistochemistry.** Immunohistochemistry results of model-related genes in colon cancer tissues and adjacent normal tissues, *P < 0.05, **P < 0.01, ***P < 0.001, ****P < 0.0001.

are considered key genes in regulating both the "establishment" and "maintenance" pathways of cellular senescence, which has been confirmed to be closely associated with tumorigenesis and progression [35]. IL-1β, as the transcriptional product of IL1B, can regulate gene expression and cytokine production, modulate cell adhesion and migration, and participate in

angiogenesis or immune responses, exerting pleiotropic effects in cancer development, migration, and metastasis [36].

Subsequently, unsupervised clustering analysis divided the tumor samples into two subtypes. Although no significant differences in survival rates were observed, and the ESTIMATE score suggested no obvious differences in immune cell infiltration levels between the two subtypes, immune cell infiltration analysis indicated variations in the proportions of specific immune cell types between the subtypes, leading to differences in immune function. This suggests that in colon cancer, different cuproptosis- and ferroptosis-related subtypes may exhibit varied responses to immunotherapy. Tailoring immunotherapy strategies based on the immune infiltration characteristics of each subtype could enhance tumor treatment sensitivity to some extent. Therefore, we further conducted univariate Cox regression and LASSO Cox regression to screen prognostic genes among the differentially expressed genes between the two subtypes and constructed a prognostic model containing seven genes: MAGEA12, C12orf56, ASRGL1, EREG, RORC, MMP10, and INHBB. MAGEA12 is a member of the human MAGE gene family, primarily encoding tumor-associated antigens recognized by various allelic forms of MHC class I molecules on the surface of tumor cells, recognized by cytotoxic T lymphocytes (CTLs) [37]. Although C12orf56 is differentially expressed in tumors and normal tissues, its mechanism of action remains unclear. ASRGL1 strongly inhibits the development of hepatocellular carcinoma by inhibiting the formation of the cyclin B/CDK1 complex, ultimately leading to the failure of transition from the G2 to M phase of the cell cycle [38]. There is also evidence suggesting that ASRGL1, as a pivotal gene related to Th17 cells, is highly correlated with the prognosis of colorectal adenocarcinoma [39]. EREG is highly expressed in various cancers, primarily activating the EGFR signaling pathway and promoting cancer progression [40]. In colorectal cancer, EREG is regulated by methylation, and its expression is associated with CIMP status and primary tumor location [41]. RORγT encoded within the RORC locus participates in coordinating the differentiation of Th17 cells [42]. Studies have indicated that Th17 cells may influence lymph node metastasis in colorectal cancer, and high RORγT/CD3 is a strong prognostic marker after colorectal cancer surgery [43]. MMP10 is a direct target of the miR-148/152 family in mammalian colon. Loss of miR-148/152 can upregulate MMP10 expression, leading to intestinal barrier disruption and promoting the occurrence and development of colitis and colitis-associated colorectal cancer [44]. INHBB is a protein-coding gene involved in the synthesis of transforming growth factor β (TGF-β) family members. Highly expressed in primary colorectal cancer, it is significantly associated with decreased patient survival rates, serving as a target of miR-34 [45, 46].

According to the predictive model, we calculated the risk scores for each patient, dividing the tumor patients into high-risk and low-risk groups. Both the total survival of the two risk groups in the training and testing cohorts showed significant differences, indicating that the predictive model can effectively predict the survival time of colon cancer patients. Analysis of differences in risk scores among different clinical subgroups showed that the risk scores were significantly different only during tumor staging, suggesting that the risk scores could reflect the degree of tumor development and infiltration to some extent. Multifactorial COX analysis of clinical information suggested that age, tumor staging, and risk scores are independent prognostic indicators of colon cancer. We constructed a nomogram based on these independent prognostic indicators, and the results of ROC and calibration curves showed a significant improvement in the prognostic performance of the nomogram, enhancing the clinical application value of the model.

The tumor microenvironment (TME), regulated by intrinsic oncogenic mechanisms and epigenetic modifications, constitutes a highly complex and heterogeneous ecosystem comprising not only tumor cells themselves but also the surrounding cellular milieu [47]. Interactions

between tumor cells and TME promote proliferation, differentiation, invasion, metastasis, and even resistance to therapeutic interventions [48]. Cell death is integral to the immune response, guiding both the immune system and tissue microenvironment to ensure tissue repair and homeostasis [49]. Immunological analysis between the two subtypes of ferroptosis and cuproptosis indicates significant differences in the infiltration of activated dendritic cells, monocytes, neutrophils, Th17, and Th2 cells, among others, as well as variations in immune functions such as antigen-presenting cell inhibition, CC chemokine receptor signaling, Cytolytic activity, and MHC class I, which may suggest differences in the tumor microenvironment between the two subtypes. Subsequent estimation scores also confirm notable differences in stromal cell composition between the two subtypes. Stromal cells, another major cellular component of TME, play a pivotal role in tumor metabolism, growth, metastasis, immune evasion, and therapy resistance [50]. Immune analysis between the two risk groups revealed that the high-risk group exhibited significantly lower levels of 19 immune cells infiltration, including activated dendritic cells (aDC), CD4 T cells, neutrophils, Th17 cells, Th2 cells, as well as decreased levels of antigen-presenting cell inhibition, CC chemokine receptor signaling, checkpoint regulation, and para-inflammation, among nine immune functions. This may indicate that lower levels of immune cell infiltration and immune function in tumors are associated with higher malignancy, facilitating tumor progression. Moreover, the estimate score suggested that the immune cell score was higher in the low-risk group compared to the high-risk group, further supporting this point. Previous studies have indicated that in several cancer immune subtypes, C3 (inflammatory) exhibits the best prognosis [27]. This contradicts our findings, possibly due to the lower abundance of C3 in colorectal cancer, resulting in false positives in the data.

GO and KEGG enrichment analysis revealed significant enrichment of differential genes between the two subtypes of ferroptosis and cuproptosis in extracellular matrix, signal receptor activator activity, receptor ligand activity, cytokine-cytokine receptor interaction, and the IL-17 signaling pathway. It is known that the extracellular matrix is a highly active part of the tumor microenvironment, influencing the behavior and metastatic potential of tumor cells. Certain molecular components of the extracellular matrix have been identified as biomarkers for colorectal cancer and targets for targeted therapy [51]. CSF1R is a novel identified dependence receptor; Ligand-free CSF1R exerts anti-tumor effects in colorectal cancer, but upon ligand binding, it promotes cancer cell proliferation [52]. It has been discovered that various chemokines can recruit immune cells to the tumor microenvironment when bound to their specific receptors, affecting angiogenesis, invasion, and metastasis in tumors [53]. Reports have also indicated that the CXCL17-GPR35 axis can activate the IL-17 signaling pathway, enhancing the viability of colorectal cancer cells, promoting their migration and invasion, and thus increasing resistance to colorectal cancer [54]. These findings suggest that these associated functions may be related to ferroptosis and cuproptosis. Current reports indicate that the Nod-like receptor (NLR) signaling pathway, fructose and mannose metabolism, and apoptosis have inhibitory effects on tumor cells, consistent with the conclusions obtained from our gene set variation analysis [55–57].

Further analysis revealed that ASRGL1 is correlated with Th17 cells and Th2 cells, while MMP10 is primarily associated with neutrophils, dendritic cells, and Th17 cells, and the risk score is correlated with Th17 cells. Th17 cells promote tissue inflammation through the production of IL-17, and it is currently believed that Th17 cells have both promoting and inhibitory effects on tumors [58], which is consistent with our findings. Th2 cells influence macrophage polarization and activation of macrophages and eosinophils, leading to the production of cytotoxic and apoptotic factors, resulting in a significant anti-tumor response [59]. Neutrophils, on the other hand, promote tumor occurrence through immunosuppression via

mediators such as ROS, inducible nitric oxide synthase, and arginase 1, and are often associated with poor prognosis in patients [60]. Additionally, dendritic cells are the most effective antigen-presenting cells capable of activating naive T cells and inducing immune memory responses in cancer, thereby promoting anti-tumor immunity [61]. This is consistent with the high expression of MMP10 predicting a better prognosis.

Currently, neoadjuvant chemotherapy has become a standard approach for the treatment of advanced colon cancer. Therefore, we calculated sensitivity scores of colon cancer patients to 198 drugs and analyzed the correlation between model genes and drug sensitivity scores. We found that as ASRGL1 expression increases, tumors become more sensitive to Dactinomycin_1911, Docetaxel_1007, Eg5_9814_1712, Epirubicin_1511, and Pevonedistat_1529. Conversely, with increased EREG expression, tumors show decreased sensitivity to Dactinomycin_1911, Epirubicin_1511, Foretinib_2040, Mitoxantrone_1810, and Sabutoclax_1849. These findings may indicate undiscovered mechanisms of these two genes in colon cancer cells. Furthermore, we compared the treatment responses to sensitive drugs between high and low-risk groups and found that the low-risk group is more sensitive to AZD8055_1059, Bortezomib_1191, Dihydrorotenone_1827, and MG-132_1862 than the high-risk group. This finding suggests that the aforementioned drugs may be combined with conventional chemotherapeutic agents to reduce drug resistance in low-risk colorectal cancer. Additionally, the impact of ASRGL1 and EREG expression on drug sensitivity in colon cancer provides valuable guidance for the discovery of novel therapeutic agents. Based on our analysis, the differential expression of MAGEA12 between tumor and normal tissues yielded inconsistent results in the TCGA and GEO cohorts. We speculate the following reasons: First, when the expression level of a gene is very low, the limitations of current technologies amplify measurement errors, ultimately leading to discrepancies between the two datasets. Second, the populations studied in the TCGA and GEO datasets differ. TCGA samples primarily represent populations from the Americas, whereas the GSE39582 cohort consists mainly of European populations, which may exhibit differences in gene expression. Third, the TCGA dataset uses RNA sequencing (RNAseq) data, while GSE39582 relies on microarray data. Compared to microarray, RNAseq is more accurate for detecting genes with low expression levels. Subsequently, we validated the expression of seven model genes in normal and colon cancer tissues using qRT-PCR. We found that ASRGL1, C12orf56, and EREG showed consistent results with both the TCGA and GEO databases, while MAGEA12 was consistent with the GEO database. In contrast, INHBB, MMP10, and RORC showed no differential expression, differing from both databases. We believe this inconsistency may be attributed to the presence of a considerable amount of normal tissue even within macroscopically visible tumor samples. During qRT-PCR, the uncertainty in sample selection increases the likelihood of obtaining a significant proportion of normal tissue, leading to biased results. Subsequent immunohistochemistry confirmed that some genes are indeed highly expressed in cancer cells, while substantial amounts of normal tissue could still be observed in tumor tissue sections. Additionally, the limited sample size may have contributed to the differences in final statistical outcomes. Our study still has limitations. First, although we constructed and validated a prognostic model based on cuproptosis and ferroptosis using public databases, additional prospective clinical data are needed to confirm the clinical significance of the model. Second, while we identified differences in immune cell infiltration components between the two cuproptosis- and ferroptosis-related subtypes, we did not explore the underlying mechanisms in detail. In particular, the regulatory roles of ASRGL1 and MMP10 in Th17 cells remain unclear and require further investigation through in vivo and in vitro experiments. Additionally, our study found that the expression of ASRGL1 and EREG can influence the sensitivity of colon cancer to certain anti-tumor drugs, and the specific mechanisms underlying these effects warrant further

exploration. Finally, the antitumor drugs identified as potentially effective against low-risk colon cancer should be further validated in clinical practice to assess their efficacy.

## Conclusions

In summary, this study is the first to identify cuproptosis- and ferroptosis-related subtypes in colon cancer. Further analysis confirmed the differences in immune cell infiltration components between the two subtypes. We constructed a prognostic model for predicting the prognosis of colon cancer patients based on 7FCSDEGs, further exploring the model's potential functions and its impact on the tumor microenvironment in colon cancer, demonstrating its prognostic value in colon cancer. Additionally, this study provides new insights into ferroptosis and cuproptosis in colon cancer, offering references for the discovery of novel biomarkers associated with ferroptosis and cuproptosis in this context.

## Supporting information

**S1 Fig. Kaplan-Meier curves for the C1 and C2 subtypes.**
(TIF)

**S2 Fig. Kaplan-Meier curves for the model-related genes.**
(TIF)

**S1 Table. Ferroptosis- and cuproptosis-related genes.**
(XLSX)

**S2 Table. 28 types of immune cells and 13 immune function-related gene sets.**
(XLSX)

**S3 Table. Samples unsuitable for inclusion in the analysis.**
(XLSX)

**S4 Table. 52 prognosis-related genes.**
(XLSX)

**S1 File. Raw data of qRT-PCR.**
(ZIP)

## Acknowledgments

We acknowledge GEO and TCGA databases for providing their platforms and data, and we would like to thank the Chongqing Key Laboratory of Molecular Oncology and Epigenetics (Chongqing, China) for providing technical guidance.

## Author Contributions

**Conceptualization:** Yinghao He, Zheng Jiang.

**Data curation:** Yinghao He, Qingshu Li.

**Formal analysis:** Yinghao He, Qingshu Li.

**Funding acquisition:** Zheng Jiang.

**Investigation:** Yinghao He, Fuqiang Liu.

**Methodology:** Yinghao He.

**Supervision:** Zheng Jiang.

**Validation:** Yinghao He, Fuqiang Liu.

**Visualization:** Yinghao He.

**Writing – original draft:** Yinghao He.

**Writing – review & editing:** Yinghao He, Fuqiang Liu, Zheng Jiang.

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
