## [Decision Letter · Decision Letter 0]

10 Sep 2024

PONE-D-24-25790Identification of cuproptosis and ferroptosis-related subtypes and development of a prognostic signature in colon cancerPLOS ONE

Dear Dr. Jiang,

Thank you for submitting your manuscript to PLOS ONE. After careful consideration, we feel that it has merit but does not fully meet PLOS ONE’s publication criteria as it currently stands. Therefore, we invite you to submit a revised version of the manuscript that addresses the points raised during the review process.

We look forward to receiving your revised manuscript.

Kind regards,

Ruo Wang

Academic Editor

PLOS ONE

Journal requirements: 1. When submitting your revision, we need you to address these additional requirements.Please ensure that your manuscript meets PLOS ONE's style requirements, including those for file naming. The PLOS ONE style templates can be found at https://journals.plos.org/plosone/s/file?id=wjVg/PLOSOne_formatting_sample_main_body.pdf and https://journals.plos.org/plosone/s/file?id=ba62/PLOSOne_formatting_sample_title_authors_affiliations.pdf. 2. Please note that PLOS ONE has specific guidelines on code sharing for submissions in which author-generated code underpins the findings in the manuscript. In these cases, we expect all author-generated code to be made available without restrictions upon publication of the work. Please review our guidelines at https://journals.plos.org/plosone/s/materials-and-software-sharing#loc-sharing-code and ensure that your code is shared in a way that follows best practice and facilitates reproducibility and reuse 3. Please include captions for your all Supporting Information files at the end of your manuscript, and update any in-text citations to match accordingly. Please see our Supporting Information guidelines for more information: http://journals.plos.org/plosone/s/supporting-information. 

Additional Editor Comments:

The reviewers made important suggestions for the manuscript and the authors should consider how to improve the value of the results of this study. In addition, the manuscript does not contain any real-world bio-experiments, which seems to limit the importance of the manuscript.

Reviewers' comments:

Reviewer's Responses to Questions

**Comments to the Author**

1. Is the manuscript technically sound, and do the data support the conclusions?

Reviewer #1: Yes

Reviewer #2: Yes

2. Has the statistical analysis been performed appropriately and rigorously? 

Reviewer #1: Yes

Reviewer #2: Yes

3. Have the authors made all data underlying the findings in their manuscript fully available?

Reviewer #1: Yes

Reviewer #2: Yes

4. Is the manuscript presented in an intelligible fashion and written in standard English?

Reviewer #1: Yes

Reviewer #2: Yes

5. Review Comments to the Author

Reviewer #1: In this manuscript entitled "Identification of cuproptosis and ferroptosis-related subtypes and development of a prognostic signature in colon cancer", the authors and others used a variety of bioinformatics techniques to establish a prognostic model for ferroptosis/copper death in colon cancer. The manuscript has strong scientific significance and advancement, but there are the following issues that need to be revised:

1. In the manuscript, the authors only used PCR technology to verify the results. Although the authors provided an effective prediction tool, I think the authors may need more experimental verification.

Therefore, I think the manuscript needs to be revised before acceptance.

Reviewer #2: The author found ferroptosis and cuproptosis-related genes in colon cancer and identified two subtypes. And through functional analysis and bioinformatics methods, the author elucidated pathway differences and biological

characteristics between these two subtypes. Then through univariate, LASSO and multivariate Cox regression, the author constructed a 7 gene prognostic model and validated it. Then the author explored the model's potential

functions and its impact on the tumor microenvironment in colon cancer. Here are the comments.

1. It is a descriptive study without any mechanisms are validated or even proposed. The value of the results of this study is limited; it provided limited help for clinical diagnosis and treatment of colon cancer.

2. This study lacks innovation. Many studies have reported the role of ferroptosis and cuproptosis-related genes in cancer through bioinformatics analysis[1,2,3,4], and there have been study reporting the relationship between ferroptosis and cuproptosis-related genes and colon cancer[5]. So The present study lacks innovation.

3. The research methods used in this study, such as Kaplan-Meier survival analysis, univariate, LASSO and multivariate Cox regression, GO and KEGG pathway enrichment analysis, nomogram, ESTIMATE, GSVA, are conventional and lack innovation.

4. For figure 11, why used qRT-PCR to detect genes expressions on colon cancer tissues and adjacent normal tissues, not IHC and western blot? In my opinion, IHC and western blot are better than qRT-PCR to detect genes expressions on colon cancer tissues and adjacent normal tissues.

5. For figure 11-O, there are 8 pairs of colon cancer tissues and adjacent normal tissues, not 10 pairs as described in the method.

[1]Li J, Zhang W, Ma X, et al. Cuproptosis/ferroptosis-related gene signature is correlated with immune infiltration and predict the prognosis for patients with breast cancer. Front Pharmacol. 2023;14:1192434.

[2]Ma Q, Hui Y, Huang BR, et al. Ferroptosis and cuproptosis prognostic signature for prediction of prognosis, immunotherapy and drug sensitivity in hepatocellular carcinoma: development and validation based on TCGA and ICGC databases. Transl Cancer Res. 2023;12(1):46-64.

[3]Li J, Liu J, Li J, et al. A risk prognostic model for patients with esophageal squamous cell carcinoma basing on cuproptosis and ferroptosis. J Cancer Res Clin Oncol. 2023;149(13):11647-11659.

[4]Luo G, Wang L, Zheng Z, Gao B, Lei C. Cuproptosis-Related Ferroptosis genes for Predicting Prognosis in kidney renal clear cell carcinoma. Eur J Med Res. 2023;28(1):176.

[5]Li Y, Wang RY, Deng YJ, Wu SH, Sun X, Mu H. Molecular characteristics, clinical significance, and cancer immune interactions of cuproptosis and ferroptosis-associated genes in colorectal cancer. Front Oncol. 2022;12:975859.

6. PLOS authors have the option to publish the peer review history of their article (what does this mean?). If published, this will include your full peer review and any attached files.

Reviewer #1: No

Reviewer #2: No

---

## [Author Response · Author response to Decision Letter 0]

13 Oct 2024

Reviewer #1: 

Comment 1: In the manuscript, the authors only used PCR technology to verify the results. Although the authors provided an effective prediction tool, I think the authors may need more experimental verification. Therefore, I think the manuscript needs to be revised before acceptance.

Response 1: Thank you very much for your valuable suggestions. Relying solely on PCR validation results has several limitations; therefore, we further conducted immunohistochemical analyses on valuable genes identified in the immune characteristic and drug sensitivity analyses. The results are presented in the revised manuscript and figures.

Reviewer #2: 

Comment 1: It is a descriptive study without any mechanisms are validated or even proposed. The value of the results of this study is limited; it provided limited help for clinical diagnosis and treatment of colon cancer.

Response 1: Thank you very much for your comments. Although our study did not validate the mechanisms of the genes, the immune characteristic and drug sensitivity analyses indicate that certain genes in the model have research value. Our future studies will further investigate these valuable genes, which may have significance for the development of targeted and chemotherapeutic agents. Additionally, our research established an effective predictive tool for estimating patient survival time, which can be utilized clinically to assess tumor malignancy.

Comment 2: This study lacks innovation. Many studies have reported the role of ferroptosis and cuproptosis-related genes in cancer through bioinformatics analysis[1,2,3,4], and there have been study reporting the relationship between ferroptosis and cuproptosis-related genes and colon cancer[5]. So The present study lacks innovation.

Response 2: We greatly appreciate your recognition of our research results. As you mentioned, the four articles [1, 2, 3, 4] respectively reported on the prognostic characteristics of cuproptosis- and ferroptosis-related genes in breast cancer, hepatocellular carcinoma, esophageal squamous cell carcinoma, and renal clear cell carcinoma. However, the focus of our study is on colon cancer. Although article [5] investigated the prognostic characteristics of cuproptosis- and ferroptosis-related genes in colorectal cancer, it differs from our study. Our research specifically focuses on colon cancer rather than colorectal cancer because we believe there are significant differences between colon cancer and rectal cancer in terms of anatomical location, clinical presentation, and case structure. For the first time, our study identifies subtypes in colon cancer by combining cuproptosis- and ferroptosis-related genes, and we developed a prognostic model based on the differential genes between the two subtypes. This approach allows us to more effectively identify potential genes related to both cuproptosis and ferroptosis. Additionally, we integrated immune cell and immune function gene sets from previous research on immune infiltration, leading to more comprehensive and precise results. Finally, while [5] used the HPA database for validation, our study employed qRT-PCR and IHC for validation. Therefore, we believe our study is more reliable compared to [5] and demonstrates a certain level of innovation.

Comment 3: The research methods used in this study, such as Kaplan-Meier survival analysis, univariate, LASSO and multivariate Cox regression, GO and KEGG pathway enrichment analysis, nomogram, ESTIMATE, GSVA, are conventional and lack innovation.

Response 3: Thank you very much for your comments. Although Kaplan-Meier survival analysis, univariate, LASSO, multivariate Cox regression and other methods are conventional, they are classic, well-established techniques that have been widely validated. While this approach may lack methodological innovation, it ensures the accuracy of the analytical results."

Comment 4:For figure 11, why used qRT-PCR to detect genes expressions on colon cancer tissues and adjacent normal tissues, not IHC and western blot? In my opinion, IHC and western blot are better than qRT-PCR to detect genes expressions on colon cancer tissues and adjacent normal tissues.

Response 4: Thank you very much for your valuable suggestions. Relying solely on qRT-PCR validation results has several limitations; therefore, we further conducted immunohistochemical analyses on valuable genes identified in the immune characteristic and drug sensitivity analyses. The results are presented in the revised manuscript and figures.

Comment 5:For figure 11-O, there are 8 pairs of colon cancer tissues and adjacent normal tissues, not 10 pairs as described in the method.

Response 5: Thank you very much for your question. Figure 11-O shows 10 pairs of colon cancer tissues and adjacent normal tissues. However, since the gene expression levels of some samples are similar, two points overlapped, making it appear as if there were only 8 pairs of tissue samples. The detailed situation can be seen in the original qRT-PCR data.

References

[1] Li J, Zhang W, Ma X, et al. Cuproptosis/ferroptosis-related gene signature is correlated with immune infiltration and predict the prognosis for patients with breast cancer. Front Pharmacol. 2023;14:1192434.

[2] Ma Q, Hui Y, Huang BR, et al. Ferroptosis and cuproptosis prognostic signature for prediction of prognosis, immunotherapy and drug sensitivity in hepatocellular carcinoma: development and validation based on TCGA and ICGC databases. Transl Cancer Res. 2023;12(1):46-64.

[3] Li J, Liu J, Li J, et al. A risk prognostic model for patients with esophageal squamous cell carcinoma basing on cuproptosis and ferroptosis. J Cancer Res Clin Oncol. 2023;149(13):11647-11659.

[4] Luo G, Wang L, Zheng Z, Gao B, Lei C. Cuproptosis-Related Ferroptosis genes for Predicting Prognosis in kidney renal clear cell carcinoma. Eur J Med Res. 2023;28(1):176.

[5] Li Y, Wang RY, Deng YJ, Wu SH, Sun X, Mu H. Molecular characteristics, clinical significance, and cancer immune interactions of cuproptosis and ferroptosis-associated genes in colorectal cancer. Front Oncol. 2022;12:975859.

---

## [Decision Letter · Decision Letter 1]

7 Nov 2024

PONE-D-24-25790R1Identification of cuproptosis and ferroptosis-related subtypes and development of a prognostic signature in colon cancerPLOS ONE

Dear Dr. Jiang,

Thank you for submitting your manuscript to PLOS ONE. After careful consideration, we feel that it has merit but does not fully meet PLOS ONE’s publication criteria as it currently stands. Therefore, we invite you to submit a revised version of the manuscript that addresses the points raised during the review process.

We look forward to receiving your revised manuscript.

Kind regards,

Ruo Wang

Academic Editor

PLOS ONE

Reviewers' comments:

Reviewer's Responses to Questions

**Comments to the Author**

1. If the authors have adequately addressed your comments raised in a previous round of review and you feel that this manuscript is now acceptable for publication, you may indicate that here to bypass the “Comments to the Author” section, enter your conflict of interest statement in the “Confidential to Editor” section, and submit your "Accept" recommendation.

Reviewer #2: (No Response)

2. Is the manuscript technically sound, and do the data support the conclusions?

Reviewer #2: Yes

3. Has the statistical analysis been performed appropriately and rigorously? 

Reviewer #2: Yes

4. Have the authors made all data underlying the findings in their manuscript fully available?

Reviewer #2: Yes

5. Is the manuscript presented in an intelligible fashion and written in standard English?

Reviewer #2: Yes

6. Review Comments to the Author

Reviewer #2: 1、Figure 11 (A-G) and (H-N) are both labeled "TCGA Expression difference," and they do not include GEO database analysis as described in the results.

2、The authors analyzed the expression differences of seven model-associated genes between tumor and normal tissues using the TCGA database, GEO database, and qRT-PCR. However, in the immunohistochemistry analysis, only 2 genes were examined. Please analyze all 7 genes using immunohistochemistry to ensure consistency with the other data.

3、The authors have already analyzed the expression differences of the seven model-associated genes between tumor and normal tissues. It is recommended to further investigate the correlation between these seven genes and patient prognosis to elucidate their impact on outcomes.

4、The introduction mentions that colorectal cancer exhibits different responses to treatment and prognoses due to the inherent heterogeneity of tumors, thus necessitating the search for new biomarkers to guide treatment. It then analyzes the cuproptosis and ferroptosis-related subtypes of colorectal cancer to clarify the tumor's intrinsic heterogeneity and guide treatment. However, there are currently no specific treatment methods targeting ferroptosis and cuproptosis in clinical practice for colorectal cancer. The author is requested to elaborate on why it is important to study cuproptosis and ferroptosis-related subtypes in colorectal cancer, the significance of this research, and its implications for future research or clinical practice.

5、The introduction points out the mutual dependence between ferroptosis and cuproptosis and then provides examples such as sorafenib and erastin, which can regulate the expression of the cuproptosis-related gene FDX1 and upregulate protein lipoylation. According to current studies, these mechanisms are related to cuproptosis. However, sorafenib and erastin primarily induce ferroptosis, and whether they can induce cuproptosis through these mechanisms remains unclear. Therefore, the mutual dependence between ferroptosis and cuproptosis cannot be definitively established. Please provide a detailed explanation of the interdependence between cuproptosis and ferroptosis, and elaborate on how these two modes of cell death interact or converge in colon cancer, thereby justifying the study’s rationale.

6、The discussion presents a broad overview of the findings, but fails to emphasize the uniqueness and specific contributions of this study to the existing body of knowledge on colon cancer. Please elaborate on the unique insights provided by this study in the field of colon cancer research and treatment, especially regarding the newly identified copper death and iron death related subtypes.

7、The discussion mentions that certain genes or data exhibit inconsistencies when compared with the TCGA and GEO databases, but it does not delve into an analysis or provide hypotheses for these differences. Please provide a detailed explanation of these data discrepancies.

8、The limitations section of the discussion is relatively brief and does not comprehensively address other potential shortcomings or areas for further research. Please offer a more in-depth exploration of the study's limitations and suggest directions for future research.

7. PLOS authors have the option to publish the peer review history of their article (what does this mean?). If published, this will include your full peer review and any attached files.

Reviewer #2: No

---

## [Author Response · Author response to Decision Letter 1]

13 Dec 2024

Point-by-point response to reviewers

We sincerely appreciate all the insightful comments and suggestions regarding our manuscript, which have greatly helped us to improve the quality of our manuscript. We have carefully considered each of the points raised and a point-by-point response to reviewers’ comments are listed below.

Reviewer #2: 

Comment 1: Figure 11 (A-G) and (H-N) are both labeled "TCGA Expression difference," and they do not include GEO database analysis as described in the results.

Response 1: Thank you for pointing out the error in our study. This was due to our carelessness, which led to incorrect figure captions. We have corrected and re-uploaded the images for Figure 11 (A-G) and (H-N).

Comment 2: The authors analyzed the expression differences of seven model-associated genes between tumor and normal tissues using the TCGA database, GEO database, and qRT-PCR. However, in the immunohistochemistry analysis, only 2 genes were examined. Please analyze all 7 genes using immunohistochemistry to ensure consistency with the other data.

Response 2: We sincerely appreciate you pointing out the shortcomings in our study. Following your suggestions, we have completed the immunohistochemical analysis of the remaining five model-related genes, and the results are presented in Figure 12.

Comment 3: The authors have already analyzed the expression differences of the seven model-associated genes between tumor and normal tissues. It is recommended to further investigate the correlation between these seven genes and patient prognosis to elucidate their impact on outcomes.

Response 3: Thank you for your suggestions regarding this study. In fact, the study has already demonstrated the correlation between model-associated genes and prognosis through univariate Cox regression analysis. To further analyze the prognostic differences between groups with high and low expression of each model-associated genes, we plotted Kaplan-Meier (KM) curves for each model-associated gene, as shown in Supplementary Figure 2. We noticed that the results for certain genes differ from those obtained through univariate Cox regression analysis. We speculate that this discrepancy may be due to methodological differences. Kaplan-Meier analysis is a non-parametric method that considers only a single grouping variable’s impact on survival time without adjusting for other potential confounders. Additionally, the choice of grouping method may also influence the results of KM analysis.

Comment 4: The introduction mentions that colorectal cancer exhibits different responses to treatment and prognoses due to the inherent heterogeneity of tumors, thus necessitating the search for new biomarkers to guide treatment. It then analyzes the cuproptosis and ferroptosis-related subtypes of colorectal cancer to clarify the tumor's intrinsic heterogeneity and guide treatment. However, there are currently no specific treatment methods targeting ferroptosis and cuproptosis in clinical practice for colorectal cancer. The author is requested to elaborate on why it is important to study cuproptosis and ferroptosis-related subtypes in colorectal cancer, the significance of this research, and its implications for future research or clinical practice.

Response 4: We sincerely appreciate your comments on this study. Based on your suggestions, we have revised the Introduction section (paragraph 4) of the manuscript. Considering that cuproptosis- and ferroptosis-related subtypes have shown immune microenvironment characteristics and prognostic differences in many other tumors[1,2], we believe it is necessary to explore their features in colon cancer. Moreover, cuproptosis and ferroptosis are currently considered to have potential in overcoming tumor resistance[3]. Therefore, this study’s preliminary exploration of the interaction between cuproptosis and ferroptosis in the immune microenvironment of colon cancer also highlights their potential in colon cancer immunotherapy. Additionally, the drug sensitivity analysis performed in this study provides more therapeutic options for resistant tumors.

Comment 5: The introduction points out the mutual dependence between ferroptosis and cuproptosis and then provides examples such as sorafenib and erastin, which can regulate the expression of the cuproptosis-related gene FDX1 and upregulate protein lipoylation. According to current studies, these mechanisms are related to cuproptosis. However, sorafenib and erastin primarily induce ferroptosis, and whether they can induce cuproptosis through these mechanisms remains unclear. Therefore, the mutual dependence between ferroptosis and cuproptosis cannot be definitively established. Please provide a detailed explanation of the interdependence between cuproptosis and ferroptosis, and elaborate on how these two modes of cell death interact or converge in colon cancer, thereby justifying the study’s rationale.

Response 5: Thank you for your suggestions. Our wording may have been unclear, so we have made corresponding revisions to the Introduction section of the manuscript in response to your concerns. Based on previous studies, it is believed that there is some crosstalk between cuproptosis and ferroptosis[4]. Mitochondria, as a critical site for both ferroptosis and cuproptosis, have the TCA cycle as their convergence point. On one hand, the mitochondrial TCA cycle and electron transport chain play a central role in initiating ferroptosis by promoting mitochondrial membrane potential hyperpolarization and the accumulation of lipid peroxides[5]. On the other hand, the essential TCA cycle protein DLAT undergoes lipoylation mediated by FDX1, and the lipoylated protein’s binding with copper triggers copper toxicity[6]. Additionally, a recent study demonstrated that ferroptosis inducers sorafenib and erastin not only induce ferroptosis but also promote cuproptosis. The underlying mechanism involves inhibiting the degradation of FDX1 mediated by mitochondrial matrix-associated proteases and suppressing intracellular GSH synthesis[7]. This suggests that intracellular GSH synthesis may act as a common mediator between ferroptosis and cuproptosis[4].

Comment 6: The discussion presents a broad overview of the findings, but fails to emphasize the uniqueness and specific contributions of this study to the existing body of knowledge on colon cancer. Please elaborate on the unique insights provided by this study in the field of colon cancer research and treatment, especially regarding the newly identified copper death and iron death related subtypes.

Response 6: Thank you for your suggestions for improvement. We have enriched the corresponding content in the Discussion section of the manuscript. First, our study clarified that although there is no prognostic difference between cuproptosis- and ferroptosis-related subtypes in colon cancer, there are significant differences in immune cell infiltration and immune function. This indicates that different cuproptosis- and ferroptosis-related subtypes may exhibit varying responses to immunotherapy in colon cancer. Individualized immunotherapy strategies tailored to the immune cell infiltration characteristics of different subtypes could potentially enhance tumor treatment sensitivity to some extent. Second, our study identified several drugs that are sensitive to colorectal cancer and exhibit higher sensitivity in low-risk groups compared to high-risk groups. These drugs could be further investigated for their potential to be combined with existing chemotherapy regimens to overcome tumor resistance.

Comment 7: The discussion mentions that certain genes or data exhibit inconsistencies when compared with the TCGA and GEO databases, but it does not delve into an analysis or provide hypotheses for these differences. Please provide a detailed explanation of these data discrepancies.

Response 7: We sincerely appreciate your valuable comments. In the Discussion section of the manuscript, we have provided a detailed explanation regarding the differences in the analysis results between the TCGA and GEO databases, as well as the discrepancies between our qRT-PCR results and the public databases. Regarding the differences between the TCGA database and the GEO database, we believe the reasons are as follows: First, when the expression level of a gene is very low, the limitations of current technologies amplify measurement errors, ultimately leading to discrepancies between the two datasets. Second, the populations studied in the TCGA and GEO datasets differ. TCGA samples primarily represent populations from the Americas, whereas the GSE39582 cohort consists mainly of European populations, which may exhibit differences in gene expression. Third, the TCGA dataset uses RNA sequencing (RNAseq) data, while GSE39582 relies on microarray data. Compared to microarray, RNAseq is more accurate for detecting genes with low expression levels. Regarding the discrepancies between the qRT-PCR results and the public database, we believe this inconsistency may be attributed to the presence of a considerable amount of normal tissue even within macroscopically visible tumor samples. During qRT-PCR, the uncertainty in sample selection increases the likelihood of obtaining a significant proportion of normal tissue, leading to biased results. Subsequent immunohistochemistry confirmed that some genes are indeed highly expressed in cancer cells, while substantial amounts of normal tissue could still be observed in tumor tissue sections. Additionally, the limited sample size may have contributed to the differences in final statistical outcomes. 

Comment 8: The limitations section of the discussion is relatively brief and does not comprehensively address other potential shortcomings or areas for further research. Please offer a more in-depth exploration of the study's limitations and suggest directions for future research.

Response 8: Thank you for your valuable comments on this study. Following your suggestions, we have conducted a more in-depth discussion of the study's limitations. First, although we constructed and validated a prognostic model based on cuproptosis and ferroptosis using public databases, additional prospective clinical data are needed to confirm the clinical significance of the model. Second, while we identified differences in immune cell infiltration components between the two cuproptosis- and ferroptosis-related subtypes, we did not explore the underlying mechanisms in detail. In particular, the regulatory roles of ASRGL1 and MMP10 in Th17 cells remain unclear and require further investigation through in vivo and in vitro experiments. Additionally, our study found that the expression of ASRGL1 and EREG can influence the sensitivity of colon cancer to certain antitumor drugs, and the specific mechanisms underlying these effects warrant further exploration. Finally, the antitumor drugs identified as potentially effective against low-risk colon cancer should be further validated in clinical practice to assess their efficacy.

References

[1] Yang BF, Ma Q, Hui Y, Gao XC, Ma DY, Li JX, et al. Identification of cuproptosis and ferroptosis-related subgroups and development of a signature for predicting prognosis and tumor microenvironment landscape in hepatocellular carcinoma. Transl Cancer Res. 2023;12(12):3327-45. Epub 20231227. doi: 10.21037/tcr-23-685. PubMed PMID: 38192999; PubMed Central PMCID: PMCPMC10774034.

[2] Li Y, Fang T, Shan W, Gao Q. Identification of a Novel Model for Predicting the Prognosis and Immune Response Based on Genes Related to Cuproptosis and Ferroptosis in Ovarian Cancer. Cancers (Basel). 2023;15(3). Epub 20230118. doi: 10.3390/cancers15030579. PubMed PMID: 36765541; PubMed Central PMCID: PMCPMC9913847.

[3] Wang JJ, Li JX, Liu J, Chan KY, Lee HS, Lin KN, et al. Interplay of Ferroptosis and Cuproptosis in Cancer: Dissecting Metal-Driven Mechanisms for Therapeutic Potentials. Cancers. 2024;16(3):19. doi: 10.3390/cancers16030512. PubMed PMID: WOS:001159184500001.

[4] Liu N, Chen MB. Crosstalk between ferroptosis and cuproptosis: From mechanism to potential clinical application. Biomed Pharmacother. 2024;171:16. doi: 10.1016/j.biopha.2023.116115. PubMed PMID: WOS:001154173100001.

[5] Gao M, Yi J, Zhu J, Minikes AM, Monian P, Thompson CB, et al. Role of Mitochondria in Ferroptosis. Mol Cell. 2019;73(2):354-63.e3. Epub 20181220. doi: 10.1016/j.molcel.2018.10.042. PubMed PMID: 30581146; PubMed Central PMCID: PMCPMC6338496.

[6] Tsvetkov P, Coy S, Petrova B, Dreishpoon M, Verma A, Abdusamad M, et al. Copper induces cell death by targeting lipoylated TCA cycle proteins. Science. 2022;375(6586):1254-+. doi: 10.1126/science.abf0529. PubMed PMID: WOS:000770386600033.

[7] Wang WK, Lu KZ, Jiang X, Wei Q, Zhu LY, Wang X, et al. Ferroptosis inducers enhanced cuproptosis induced by copper ionophores in primary liver cancer. J Exp Clin Cancer Res. 2023;42(1):12. doi: 10.1186/s13046-023-02720-2. PubMed PMID: WOS:001000971000001.

---

## [Decision Letter · Decision Letter 2]

5 Jan 2025

Identification of cuproptosis and ferroptosis-related subtypes and development of a prognostic signature in colon cancer

PONE-D-24-25790R2

Dear Dr. Jiang,

We’re pleased to inform you that your manuscript has been judged scientifically suitable for publication and will be formally accepted for publication once it meets all outstanding technical requirements.

Kind regards,

Ruo Wang

Academic Editor

PLOS ONE

Additional Editor Comments (optional):

Reviewers' comments:

Reviewer's Responses to Questions

**Comments to the Author**

1. If the authors have adequately addressed your comments raised in a previous round of review and you feel that this manuscript is now acceptable for publication, you may indicate that here to bypass the “Comments to the Author” section, enter your conflict of interest statement in the “Confidential to Editor” section, and submit your "Accept" recommendation.

Reviewer #2: All comments have been addressed

2. Is the manuscript technically sound, and do the data support the conclusions?

Reviewer #2: Yes

3. Has the statistical analysis been performed appropriately and rigorously? 

Reviewer #2: Yes

4. Have the authors made all data underlying the findings in their manuscript fully available?

Reviewer #2: Yes

5. Is the manuscript presented in an intelligible fashion and written in standard English?

Reviewer #2: Yes

6. Review Comments to the Author

Reviewer #2: (No Response)

7. PLOS authors have the option to publish the peer review history of their article (what does this mean?). If published, this will include your full peer review and any attached files.

Reviewer #2: No

---

## [Editor Report · Acceptance letter]

22 Jan 2025

PONE-D-24-25790R2 

PLOS ONE

Dear Dr. Jiang, 

I'm pleased to inform you that your manuscript has been deemed suitable for publication in PLOS ONE. Congratulations! Your manuscript is now being handed over to our production team.

Kind regards, 

on behalf of

Dr. Ruo Wang 

Academic Editor

PLOS ONE